# Modeling CRISPR gene drives for suppression of invasive rodents using a supervised machine learning framework

**Samuel E. Champer**[1⊙], **Nathan Oakes**[1⊙], **Ronin Sharma**[1], **Pablo García-Díaz**[2], **Jackson Champer**[1], **Philipp W. Messer**[1]*

**1** Department of Computational Biology, Cornell University, Ithaca, New York, United States of America,
**2** Manaaki Whenua–Landcare Research, Lincoln, New Zealand and School of Biological Sciences, University of Aberdeen, Aberdeen, United Kingdom

⊙ These authors contributed equally to this work.
* messer@cornell.edu

**Data Availability Statement:** Data and Model Availability The data generated for this paper, the population model, and the GP models are available at https://github.com/MesserLab/

## Abstract

Invasive rodent populations pose a threat to biodiversity across the globe. When confronted with these invaders, native species that evolved independently are often defenseless. CRISPR gene drive systems could provide a solution to this problem by spreading transgenes among invaders that induce population collapse, and could be deployed even where traditional control methods are impractical or prohibitively expensive. Here, we develop a high-fidelity model of an island population of invasive rodents that includes three types of suppression gene drive systems. The individual-based model is spatially explicit, allows for overlapping generations and a fluctuating population size, and includes variables for drive fitness, efficiency, resistance allele formation rate, as well as a variety of ecological parameters. The computational burden of evaluating a model with such a high number of parameters presents a substantial barrier to a comprehensive understanding of its outcome space. We therefore accompany our population model with a meta-model that utilizes supervised machine learning to approximate the outcome space of the underlying model with a high degree of accuracy. This enables us to conduct an exhaustive inquiry of the population model, including variance-based sensitivity analyses using tens of millions of evaluations. Our results suggest that sufficiently capable gene drive systems have the potential to eliminate island populations of rodents under a wide range of demographic assumptions, though only if resistance can be kept to a minimal level. This study highlights the power of supervised machine learning to identify the key parameters and processes that determine the population dynamics of a complex evolutionary system.

## Author summary

Invasive rodents can devastate biodiversity on small islands. This is because many types of plants and animals that evolved on such islands have no natural defense mechanisms against a rapidly spreading new invader. Gene drive is a promising new technology that,

GeneDriveForSuppressionOfInvasiveRodents.
Among the available files is a Jupyter notebook that
loads pre-trained GP models and which can be
used to generate heatmap graphs such as those
presented in this paper. A series of animated
heatmap plots wherein three parameters are varied
at a time is also available in the GitHub repo. The
SLiM simulation software used in the project is
available at https://github.com/MesserLab/SLiM.

**Funding:** This study was supported by funding
from New Zealand's Predator Free 2050 program
under Predator Free 2050 Ltd. award SS/05/01 to
PWM, and from National Institutes of Health award
R01GM127418 to PWM. PG-D received funding
from the New Zealand BioHeritage National Science
Challenge (contract 1617-28-033 A to Manaaki
Whenua – Landcare Research) and from Natural
Environment Research Council grant NE/S011641/
1 under the Newton Latam programme. The
funders had no role in study design, data collection
and analysis, decision to publish, or preparation of
the manuscript.

**Competing interests:** The authors have declared
that no competing interests exist.

among other applications, may help control invasive rodent populations. A well-designed
gene drive system could spread an engineered gene throughout a rodent population and
eventually cause the population to collapse. We developed a detailed computational
model of the release of a suppression gene drive into an island rat population and demon-
strate that an efficient enough drive could indeed eradicate such a population within sev-
eral years. To assist with a detailed analysis of our model, which involves various
ecological and genetic parameters, we also developed a machine learning model to match
the outcomes of the underlying population model. After sufficient training, this machine
learning model is a close match to the underlying model, but runs thousands of times
faster, thereby allowing for a much more detailed analysis of the behavior of the model.
We believe that this new technique could be applied to the study of many other complex
evolutionary systems.

## Introduction

Global commerce webs and rapid human migration patterns that have arisen over the last sev-
eral hundred years have resulted in the spread of various invasive species across the globe [1].
Unfortunately, some ecosystems are highly vulnerable to disruption by these invaders, result-
ing in severe ecological consequences to endemic species. Rodents such as rats can be particu-
larly damaging when introduced to remote islands, where they may find themselves
completely without predators. These small, resourceful mammals can find ample food sources
in the form of flora and fauna that evolved no defenses against such enemies. To date, humans
have introduced rats to more than 80% of the planet's island groups [2].

Local eradication of invasive rat species is a critical conservation strategy on islands where
the invaders threaten endemic species. Eradication efforts on small and medium-sized islands
have successfully protected endangered species from extinction [3]. These efforts tend to con-
sist of combinations of hunting, trapping, and poisoning. However, it is not always practical to
scale up such strategies on larger islands. If even a small remnant of invaders survives, the pop-
ulation can rapidly bounce back after control efforts cease. This means that impactful results
cannot be achieved unless these strategies are applied continuously [4]. Such methods are also
controversial due to the risks of water contamination, suffering of the targeted species, and
inadvertent exposure and lethality to non-target species, especially native birds and livestock
[5,6]. Hence, there is a clear and urgent need for new approaches to combat invasive rodent
species in order to preserve native biological diversity.

The recent development of flexible CRISPR gene editing technology has prompted the con-
sideration of "gene drives" as potential tools to control invasive pest populations. A gene drive
is an allele that biases its genetic inheritance such that the drive allele is transmitted at higher
than Mendelian ratios [7–13]. While natural gene drives have been observed [14], CRISPR
technology has enabled the creation of artificially engineered gene drive constructs. One
prominent form of gene drive is the "homing drive", a system in which the drive allele carries
an endonuclease gene such as Cas9 that targets and cleaves a genomic site dictated by a guide
RNA (gRNA) in germline cells. After cleaving the target site, the cell repairs the double strand
break via homology-directed repair, a process that results in the drive allele being copied onto
the previously cleaved chromosome. If such a gene drive functions with 100% efficiency,
drive/wild-type heterozygotes only produce drive-carrying gametes. As a result, the drive can
quickly spread through a population, even if it imposes a fitness cost. The design of such

systems has been successfully demonstrated in a variety of organisms, including yeast [15–18], flies [19–26], mosquitoes [27–30], and most recently, mice [31].

Among other potential uses, gene drives can be designed to reduce or even outright eliminate a target population. Such an approach could potentially rid islands of invasive rat populations even where other control strategies cannot. For example, a recent study demonstrated the first successful elimination of a cage population of *Anopheles gambiae* mosquitoes using a homing drive that targets an essential female fertility gene such that drive homozygous females are sterile while heterozygotes can still reproduce [32]. The drive was able to spread through the population, and once it was present at a high enough frequency, the population quickly collapsed.

The potential use of gene drives for the control of invasive species has sparked intense controversy among scientists, regulators, politicians, and the public [33–36]. One key concern is whether a gene drive release can be ensured to achieve the desired outcome and avoid any unintended consequences, such as the spread of the drive beyond the intended target population or the evolution of resistance alleles against the drive. To enable an informed discussion of this issue, it is critical that accurate models be developed to predict the expected dynamics and outcome of a gene drive release. These models must account for the fact that real-world populations can differ profoundly from the small populations typically studied in laboratory experiments. For example, spatial population structure could have a major impact on the success or failure of a gene drive in a real-world population [37–43]. Large natural populations could also provide a higher chance for resistance to evolve against a drive, which could ultimately thwart its spread [22,27,44,45].

Numerous modeling approaches have been utilized to this end, which can be loosely categorized by the degree of abstraction present in the model. A decrease in the degree of abstraction in a model is generally accompanied by an increase in computational complexity. Models based on differential equations present a high degree of abstraction, tend to have relatively few parameters, and can usually be rapidly evaluated at any given point in the parameter space [39,46–49]. Wright-Fisher models are less abstract while remaining fairly computationally tractable, simulating a population as a panmictic collection of individuals [12,50–52], though the cost of evaluation often increases proportionately with the number of individuals. One notable abstraction made by these models is the assumption of discrete, non-overlapping generations [53,54]. Abandoning this abstraction in favor of simulating smaller time steps represents a further increase in computational complexity. Spatially explicit models [37,40,41,55,56] represent yet another decrease in abstraction–one that typically comes at the cost of a significant increase in computational overhead. The evaluation time of a spatial model in which individuals interact with one another tends to scale with the square of the number of individuals. Not only does a less abstract model therefore take longer to evaluate at any given point in its parameter space, such models tend to also feature an increased number of potentially variable parameters. For instance, a panmictic model need not include any parameters related to migration.

It is reasonable to make abstractions when doing so does not substantially impact the results of a simulation. In the case of gene drive, however, mounting evidence suggests that many of these abstractions can substantially alter the outcomes that models predict [37,39,40]. Relying on models that make these abstractions could therefore result in an imperfect impression of the conditions under which a drive could be successful in an actual release. Several studies have highlighted the importance of using more realistic population models when assessing the expected outcome of a suppression drive. For example, suppression drives with parameters that invariably achieve eradication in a panmictic population model can sometimes fail in a model with a spatially structured population due to complex metapopulation dynamics

wherein local population collapse is followed by recolonization by wild-type individuals [37,39,57]. The precise nature of the dynamics that play out depends on factors such as migration behavior, the rate at which wild-type populations can rebound after recolonization of previously cleared areas, seasonality, and other species-specific characteristics.

In this study, we develop a spatially explicit model of a gene drive release into an island population of rats with the goal of assessing the ability of the drive to eliminate such a population under a wide range of demographic and ecological assumptions. Our model seeks to accurately simulate life cycles and dispersal behavior within a rat population with overlapping generations, and includes a genetic component simulating three different types of suppression gene drives with configurable fitness costs, efficiency, and resistance rates. The first drive that we model is a homing drive which targets and disrupts an essential but haplosufficient female fertility gene. The second is a homing drive that disrupts a haplosufficient gene essential for zygote viability. The third is a non-homing Y-shredder located on the X-chromosome that biases the population towards females by eliminating Y-chromosome-bearing gametes in males [50].

Our model includes eight variable demographic and ecological parameters and five variable drive-related parameters. An exhaustive exploration of the parameter space of a model with this number of parameters would pose a substantial undertaking. One possible approach to ameliorate this problem is to rely on human intuition to choose which parameters to vary and which to leave at fixed values. Yet, both the selection of which parameters to vary and the selection of values at which to fix other parameters are serious points of potential failure. Important interactions between parameters could be overlooked, and decisions about fixed parameters can be invalidated by new ecological data. A model that fixes species characteristics at incorrect values runs the risk of being a less accurate predictive tool than a more abstract model that doesn't even simulate those characteristics in the first place.

Another option is to perform a variance-based sensitivity analysis [58,59]. Such an analysis starts with a set of input parameters sampled from a space-filling distribution and then uses model outputs to induce the relative contribution of each parameter to the variability of the output. However, the number of inputs required by such an analysis grows rapidly with the number of parameters of the model, and obtaining reasonable confidence bounds on such an analysis can therefore require the evaluation of a prohibitive number of simulations.

In order to enable a thorough probing of the parameter space without fixing any parameters, we use a machine learning interpolation algorithm to create a meta-model that can predict the outcome of our population model with a high degree of accuracy at a small fraction of the computational cost.

We implement this meta-model as a Gaussian process (GP) model. In addition to acting as an interpolator between sparsely sampled data points, a key feature of a GP is that it does not just make predictions, but also includes uncertainty information regarding its predictions [60]. Because a GP is aware of its own uncertainty, it can be adaptively trained on new data points sampled from regions of greatest uncertainty. After several iterations of adaptive sampling, our GP meta-model performs with a high degree of accuracy and several orders of magnitude more rapidly than the underlying population model, enabling high quality sensitivity analyses. These analyses can provide us with an in-depth understanding of the outcome space, including the identification of complex interactions among different parameters. We believe that machine learning techniques similar to those used in this study may help shed new light on the dynamics of other complex ecological systems by unlocking hitherto impractical or impossible analyses.

## Methods

### I. Population model

We implemented an individual-based population model designed to describe an island population of rats with characteristics similar to black rats (*Rattus rattus*) or brown rats (*Rattus norvegicus*) [61–64]. The model is implemented in the SLiM forward-in-time population genetic simulation framework (version 3.3.1) [65]. The simulation takes place across a homogenous continuous space, modeled as a square area with a side length configurable to between one and five kilometers. The exact number of individuals present in the simulation is not fixed and tends to fluctuate stochastically around an expected carrying capacity. For a list of demographic parameters, see Table 1.

Time steps in the simulation are equivalent to one breeding cycle, which in reality corresponds to an average of two to three months, but can range from approximately one to six months depending on the species, availability of resources, as well as the climate and time of year. With black rats and brown rats having average lifespans of one and two years respectively, it is expected that individuals in the model will be able to survive for several breeding cycles. At the beginning of each time step, adults mate and produce offspring. This is followed by mortality calculation and migration, after which time advances to the next step.

The simulation is initialized by randomly distributing a number of individuals equal to the expected carrying capacity throughout the space. The population is then allowed to equilibrate for 20 time steps, after which gene drive carrying individuals are introduced to the population. The simulation is then run for an additional 500 time steps or until the population is eliminated. The choice of 500 time steps was made because this represents a highly permissive timeline that provides almost all gene drives that are capable of suppressing the population a sufficient amount of time to do so. A drive that is not able to achieve success within this timespan would have little practical value in a real-world application, as this equates to over 80 years even assuming a time step interval averaging only two months.

**Reproduction.**   Each adult female in the population randomly selects an adult male from within her home range, which is defined as a circle with a radius equal to the interaction distance parameter. A female does not reproduce if there are no adult males within this area. Individuals are defined as adults starting the time step after they are born. A given male can be randomly selected by any number of females. The number of offspring produced by a pairing is drawn from a Poisson distribution with a mean defined by the litter size parameter. Newly generated offspring are each placed in a position that is offset from their mother in a random direction and a distance drawn from an exponential distribution with a mean equal to the average dispersal distance parameter, representing offspring leaving the nest after they are

**Table 1. Demographic/Ecological Parameters.**

| Demographic/ecological Parameters | Minimum | Default | Maximum |
|---|---|---|---|
| Density (individuals / km$^2$) | 600 | 1000 | 1500 |
| Island side length (km) | 1 | 2 | 5 |
| Interaction distance (m) | 60 | 75 | 300 |
| Average dispersal distance (m) | 25 | 250 | 1000 |
| Survival rate | 0.7 | 0.9 | 0.95 |
| Litter size | 2 | 4 | 8 |
| Migrant frequency | 0 | 0.1 | 0.5 |
| Adult dispersal distance multiplier | 1 | 2 | 5 |

weaned. These default reproduction rules are subject to change by gene drive mechanics, as described below.

**Migration.**  In addition to the offspring dispersing from their mothers, other individuals may migrate as well. Every time step, each adult has a probability to migrate equal to the migrant frequency parameter, representing the possibility that individuals will leave their nests in search of a new home area. Migrants move a distance equal to a draw from an exponential distribution with a mean defined by the average dispersal distance parameter multiplied by the adult dispersal distance multiplier parameter. This represents the ability of fully grown individuals to potentially cover more ground than young individuals. An exponential distribution was chosen because data indicates that motivated rats are capable of traveling vast distances under some conditions [66,67]. An exponential distribution should mirror this behavior. In both migrant movement and offspring placement, if a location is drawn that falls outside the bounds of the simulation, a new location is drawn (known as "reprising" boundaries) [68].

**Survival rate.**  Non-migrant adults experience a flat mortality rate equal to one minus the survival rate parameter. Juveniles and migrant adults are less likely to survive, accounting for higher rates of mortality among newborns, as well as competition for territory and resources in a new area. This increased mortality rate grows with the local density of individuals. Individuals experience competition from every other individual within a radius defined by the interaction distance parameter. The strength of competition exerted by any given individual is a function of the distance to that individual, and is defined by a Gaussian curve with a maximum value of one and a standard deviation ($\sigma$) of one-third of the interaction distance parameter, i.e.:

$$competiton = e^{\frac{-distance^2}{2\sigma^2}}$$

This results in closer-together individuals competing more intensely than further-apart individuals, which should prevent over-clustering within the population [69,70]. Individuals with interaction areas that extend beyond the boundaries of the simulation tend to have fewer competitors, which could result in clustering near the edges of the island. In order to prevent such edge clustering, the total competition these individuals experience is scaled to compensate for the missing area. The total competition experienced by an individual is then scaled by a density tuning coefficient, described below. The final survival rate of juveniles and migrants is given by the survival rate parameter minus the scaled total of competition.

**Carrying capacity.**  In our model, each female can reproduce a non-deterministic number of times, and individuals survive at different rates if they are migrant or non-migrant. Because of this complex behavior, the population size is regulated via a "density tuning coefficient" which is calculated at the outset of the simulation, and which calibrates the capacity of the system to approximately the amount expected given the specified density and island side length parameters.

The density tuning coefficient is calculated by approximating the population size $N$ over time as a recurrence relation. Loosely:

$$N(next\ time\ step) = non\text{-}migrants * s_0 + (migrants + newborns) * s_d$$

where $s_0$ is the default survival rate, and $s_d$ is the reduced survival rate of individuals experiencing density dependent competition. More specifically, if a population at time $t$ has size $N(t)$, given that the number of newborns is equal to half the population size (the number of females) multiplied by the litter size parameter $L$, and given the migrant frequency $M$, we have:

$$N(t+1) = N(t) * (1-M) * s_0 + N(t) * (L/2 + M) * s_d$$

The survival rate of individuals experiencing density dependent competition is defined as:

$$s_d = s_0 - density\ tuning\ factor * avg\ competition$$

Given the Gaussian competition function, the average expected competition can be solved for. Average competition is found by taking the integral of the density competition function, which is then integrated over the area of competition. This results in:

$$avg\ competition = 2\pi\sigma^2 * N(t) * (1 + L/2)$$

Finally, these equations can be solved for the required density tuning coefficient by setting $N(t)$ and $N(t+1)$ to both equal the desired carrying capacity of the system (as defined by the density parameter and the island size parameter).

This recurrence relationship is a good approximation of the actual population. However, variations in population density that can develop across the landscape, along with other stochastic factors, mean that the actual population can have a slightly different trajectory. The approximation also does not take into account the possibility of females not being able to find mates, though this should be a rare occurrence in an undisturbed population. The result is that the actual carrying capacity of the simulation can vary from the expected carrying capacity by a few percent, depending on the exact parameters of the simulation.

**Choice of parameter values for the population model.** The ranges selected for the demographic parameters were chosen to represent a wide variety of both black and brown rat populations in New Zealand (Table 1). These ranges were selected after carrying out a systematic literature review to gather information on demographic characteristics of wild populations [71]. This report is available at https://datastore.landcareresearch.co.nz/dataset/rodent-review-datasets. In cases where the real-life parameters are more difficult to accurately determine (e.g., migrant frequency), we opted to explore wide intervals to ensure that the vast majority of real-life populations of both species of rat are encompassed by our parameter space.

**Gene drive model.** Three gene drive strategies are modeled: (i) a homing drive targeting an essential but haplosufficient female fertility gene, (ii) a homing drive targeting an essential but haplosufficient viability gene, and (iii) a non-homing Y-shredder located on the X-chromosome. For a list of drive-related parameters, see Table 2.

The gene drive release is accomplished by randomly selecting a percentage of individuals in the simulation, as defined by the release percentage parameter, and converting them to drive/wild-type heterozygotes. These individuals immediately experience density dependent mortality as if they were migrants needing to establish themselves in new territory.

The mechanism of the two homing drives is that, when generating offspring, there is a possibility that gametes from heterozygote parents have been converted from wild-type to drive. The first step is the possibility of resistance allele formation, which occurs at a rate controlled by the resistance rate parameter. Resistance alleles can be of type "r1" or of type "r2". Resistance alleles of type r1 preserve the function of the targeted gene, while r2 alleles disrupt the target gene, just as the drive itself does. A portion of the resistance alleles formed are generated

**Table 2. Gene Drive Parameters.**

| Parameter | Minimum | Default | Maximum |
|---|---|---|---|
| Release percentage | 0.01 | 0.1 | 0.5 |
| Drive fitness | 0.5 or 0.75 | 1 | 1 |
| Drive efficiency | 0.5 or 0.75 | N/A (variable) | 1 |
| Resistance rate | 0 | 0 | 0.1 |
| Relative R1 resistance rate | 0 | 0 | 0.02 |

as type r1 alleles, as specified by the r1 resistance rate parameter. Otherwise, the allele will be of type r2. Generally, r2 alleles are much more common since the end-joining repair process that creates resistance alleles usually introduces a frameshift mutation or otherwise sufficiently changes the protein structure to prevent its effective function [11,19,20,22–24,28,31]. If resistance does not form, there is a chance that the gene drive is copied via homology directed repair, converting the chromosome to a drive chromosome. This likelihood is defined by the drive efficiency parameter. Note that this means a drive will convert fewer wild-type alleles to drive alleles if the resistance rate is higher, even given the same drive efficiency parameter. If neither resistance nor drive conversion occurs, the offspring receives the unaltered wild-type allele. Aside from these homing mechanics, the drive targeting a haplosufficient female fertility gene renders female drive homozygotes completely sterile. The drive targeting the haplosufficient viability gene results in drive homozygotes being immediately removed when generated. In both drives, r2 alleles impact fertility or viability as if they were drive alleles.

In contrast with the other two drives, the Y-shredder is not a homing drive (though homing sex-chromosome shredder drives have been proposed and constructed [50,72–74]). The drive is located on the X-chromosome and shreds the Y-chromosome in germline cells. This results in drive carrying males producing offspring at a biased ratio determined by the drive efficiency parameter. For example, a male with a drive efficiency of 80% has offspring at a ratio of nine females to one (we define drive efficiency as the fraction of offspring that are generated as females instead of males; 50% would normally be generated as male; this is multiplied by the 80% drive efficiency, resulting in the total percent of offspring that are generated as female being increased from 50% to 90%). The biased sex of the offspring is also the mechanism of drive spread since the daughters of drive carrying males are all drive carriers. Resistance was not simulated against the Y-shredder because the drive does not rely on homology directed repair and could likely target a large number of sites simultaneously with several gRNAs [45].

Aside from the drive mechanics themselves, the drive allele can also be configured to have a fitness cost via a drive fitness parameter. For simplicity, the drive is considered to be dominant for purposes of fitness impact, which is thus modeled as an equal cost for both homozygotes and heterozygotes. The fitness cost of the drive is implemented as a flat survival rate multiplier that is applied during each time step and has no other effect (e.g., on fecundity) except as defined by the drive mechanics.

For the two homing drives, we explored the parameter space separately with resistance simulated as well as with resistance fixed at zero. When exploring the parameter space without resistance, the minimum values for drive fitness and drive efficiency were set at 0.5. When exploring the parameter space with resistance, as well as the parameter space for the Y-shredder, the minimum values for drive fitness and drive efficiency were set at 0.75.

The data collected from each simulation includes the following: the actual population capacity (as averaged over the ten time steps before the drive is released), the frequency of the drive allele at the end of the simulation, the frequency of resistance alleles at the end of the simulation, the minimum population size from throughout the simulation, the population size at the end of the simulation, and the time step at which the population was eliminated if it was eliminated.

In the context of a suppression gene drive release, "failure" can describe several qualitatively different scenarios. These include drive loss (either due to fitness effects or competition preventing the drive from spreading), local collapse before the drive can sufficiently spread, or equilibrium outcomes wherein the drive does not have sufficient suppressive power, but merely reduces the population to a new equilibrium size [12,37,46,75]. Another possible scenario is "chasing" dynamics wherein the drive recurrently eliminates the population from local areas, which are then recolonized by wild-type individuals entering from other areas [37]. A

final possible failure outcome is a selective sweep of "r1" resistance alleles, which preserve the function of the target gene. These alleles can spread rapidly as drive and wild-type alleles are eliminated [23]. By contrast, drive success generally takes only two forms. The drive either steadily increases until the population collapses, or the population might be eliminated only after a period of fluctuation or chasing dynamics in which the drive eventually becomes too pervasive within the remaining population, which then collapses. While it is possible to classify each of these success and failure scenarios differently [37], the transitions between each state can be quite ambiguous, and we therefore focused our analysis on simply determining if the drive completely eliminated the population or if it failed to do so.

## II. Gaussian process framework

We trained a machine-learning based meta-model on our population model in order to enable a more computationally efficient exploration of the outcome space of the model. The meta-model is implemented as a Gaussian process (GP). A GP model can act as a regressor between sparsely sampled data points and also includes a confidence interval for each point that it predicts. A GP defines a probability distribution over infinitely many possible functions. This prior distribution is updated as the model is conditioned on training data and undergoes iterative marginalization, resulting in a predictive posterior distribution over the function set that can then be sampled to yield a predicted mean and a confidence interval [60]. Our GP models were constructed using the GPyTorch library for Python, which is built on the PyTorch machine learning framework and can utilize GPU resources for parallel processing through Nvidia's CUDA platform (GPyTorch version 1.0.1, PyTorch version 1.4.0, CUDA version 10.1, Python version 3.7.6) [76–79].

The predictions that a GP makes are substantially impacted by its covariance function, known as a kernel [60,80,81]. We chose the Matérn kernel with a smoothness parameter $v = 0.5$, which is equivalent to the absolute exponential kernel [80]. The function approximations produced by this kernel are less smooth than those produced by many others, making this a kernel suitable for approximating functions with rapid changes in slope.

**Model output.**   For each possible input point in the parameter space, the population model yields many outputs that we may be interested in. While a single GP model can be trained to produce multiple outputs, we chose to train GPs on only one output at a time for the sake of simplicity. We selected two different outputs that seemed most descriptive of whether the drive succeeded or failed at a given point in the parameter space and trained a GP separately on each.

The first output is the percent of simulations at each point that resulted in complete suppression, as determined from 20 replicates at each simulated point (we will refer to this as the "suppression rate" model). While this constitutes an intuitive choice, an initial assessment of our data gave us some concern that this GP might have trouble drawing good inferences. Specifically, we observed that our data was largely binary–in the data sets for the viability targeting homing drive and the Y-shredder, less than 1% of parameter points resulted in a suppression rate other than 0% or 100% (with the female fertility target, this was about 5% of points). We further observed that transitions between drive success and failure can be extremely sudden–a parameter point at which the drive fails in every simulation might instead result in success in every simulation if the drive efficiency is increased by only two or three percent (this was our motivation for selecting the Matérn kernel).

We hypothesized that this feature of the outcome space could cause the GP model to make suboptimal inferences. Consider, for instance, a drive that succeeds about half of the time when drive efficiency is 0.8 (with all other parameters fixed at a set of values $X$); when the drive

efficiency is 0.81 or higher, the drive always succeeds; when the drive efficiency is 0.79 or lower, the drive always fails. Within this context, suppose we have a sparse training set with a point at (0.5, $X$), and a second point at (1.0, $X$). In the absence of any other information, a simple interpolator should assume a transition from success to failure at 0.75. Yet if the second point were instead at (0.9, $X$), an interpolator will assume a transition point at 0.7. This problem could result in the GP requiring much more data in order to perform accurately. A training set with (0.5, $X$) and (0.75, $X$) might be even more harmful–with these points, the marginal effects of the parameter on the population model are completely lost, and the interpolator learns nothing (or worse, learns that drive efficiency *does* nothing). A GP model is more nuanced than a linear interpolation, but these issues could nonetheless impact the model's ability to fit the underlying function.

With these potential issues in mind, we trained a second GP model on a composite value gathered from our simulations (the "composite" model). This value consists of different information depending on whether or not suppression occurred. When suppression occurs, this value corresponds to the speed at which suppression occurred; specifically, the time of suppression divided by the maximum allowed number of time steps in the simulation (500). When suppression does not occur, this value corresponds to how much of an impact, if any, the drive had on the population size. Specifically, the composite value is calculated as:

$$f(X) = \begin{cases} 1 - \dfrac{time\ of\ suppression}{500}, & \text{Drive success} \\[2ex] -\dfrac{min\ pop\ size}{capacity\ pop\ size}, & \text{Drive failure} \end{cases}$$

This function is continuous between values where the drive suppresses and fails to suppress, with values near one representing drives that rapidly suppress the population, values near negative one representing drives that do nothing to the population, and with a value of zero representing the weakest possible successful drive (which manages to suppress the population only at the very last of the allotted 500 time steps). As before, each data point consists of an average from 20 replicates.

The composite model could potentially alleviate some of the problems the suppression model can face with smaller training sets by "widening" the intervals over which the outcome of the drive transitions from failure to success. However, the predicted output values could in turn also be less accurate in these areas. For example, when data points are averages of some simulations wherein suppression occurred and some simulations wherein it did not, the composite value might not line up perfectly with expectations of the rate of suppression. Consider, for instance, a data point consisting of 20 runs, eight of which caused suppression at time step 400, and twelve of which caused the population to decrease to ten percent of carrying capacity. This results in a composite value of 0.02. A composite value greater than zero should indicate a point at which we expect the drive to usually succeed, which does not completely square with the fact that only 2/5 of simulations resulted in elimination of the population in this example.

We decided to implement both types of models (suppression rate and composite), hypothesizing that the strengths and weaknesses of these two training approaches may be complementary, with the "suppression rate" GP performing inference poorly without a larger data set, but perhaps being better able to describe the areas of transition from drive success to drive failure after sufficient data is gathered.

**GP assessment.** We evaluated the accuracy of our GP models by assessing root-mean-square error (RMSE) between actual output generated by the population model and the output predicted by the GP models, as well as by measuring precision and recall. For the suppression

rate model, the RMSE was calculated between model predictions and the actual observed suppression rate. For the composite model, model predictions were scaled from [−1,1] to [0,1], after which those predictions were also compared to the observed suppression rate. While RMSE is a standard choice when assessing the fit of an interpolator, the measurement is not always a good indicator of model quality if a data set is unbalanced. For example, if the data consists of thousands of near-zero values and only a handful of successes, a model that simply always predicts zero has a very low RMSE despite not being a particularly useful model.

For this reason, we also assessed our GP models by measuring their precision and recall. Precision measures the proportion of predicted positives that are true positives (*true positives / all positive predictions*); recall measures the proportion of actual positives that the model identifies (*true positives/actual positives*). We define "actual positives" as any point in the data that resulted in suppression in at least 50% of simulations. We then map our GP models to a binary classification, with the suppression rate model considered to predict a positive when the model predicts 0.5 or greater and the composite model considered to predict a positive when the model predicts 0 or greater. As mentioned, the vast majority of the parameter space consists of points where suppression occurs in 0% or 100% of simulations, so very little fidelity is lost by this mapping to a binary classification. This method of assessment is robust to imbalanced data sets.

For the testing data sets, we generated 10000 parameter points from a Latin hypercube sampling of the parameter space for each drive. This sampling ensures that a set of points is fairly representative of the possible variability in the space [82]. In the test sets for the Y-shredder and the homing drives with resistance, there were very few points where the drive was successful, so we supplemented these testing sets with an additional 3000 Latin hypercube sampled points with drive fitness and efficiency of 0.9 and greater in order to ensure that there were sufficiently many successes to be able to accurately evaluate the model.

Because the outcome of our population model is stochastic, each point in our data sets was simulated 20 times. An alternative approach would have been to simply sample 20 times as many points. However, the computational complexity of training the GP models scales with the square of the number of data points [76], so we elected for a smaller data set wherein each point has more "weight" in terms of how much information it provides. We created five separate groups of data sets: one each for the female fertility targeting homing drive with and without resistance, one each for the viability gene targeting homing drive with and without resistance, and one for the Y-shredder. The data sets for the homing drives without resistance have lower minimum drive fitness and drive efficiency values (0.5 instead of 0.75) and will also serve to cross-validate the GP models for the same drives with the added resistance parameters.

**Training procedure.** GP training is an iterative process wherein the model is repeatedly tuned and optimized (Fig 1). Too little training can result in the model failing to draw inferences that it otherwise could, yet performing too many training iterations on the same data set can result in overfitting and a gradual decline in model quality. Furthermore, training is stochastic, so the amount of training required to get the best possible model can vary. In practice, it can be difficult to identify the sweet spot without evaluating the model against a testing data set. Yet doing so with our actual test sets would result in a GP that is specifically tuned to fit the data set that is being used to evaluate it. To circumvent this issue, we generated a validation data set to use as a testing set to evaluate interim models. This validation data set also consists of 10000 Latin hypercube sampled points for each drive, further bolstered with some additional points sampled from areas of the model where we expect the drives to perform well.

To train an initial instantiation of the GP models for each drive, we evaluated 1000 parameter points from a Latin hypercube sampling of the parameter space. These models (as well as all subsequent models) were trained for 30000 iterations, with the model being saved every

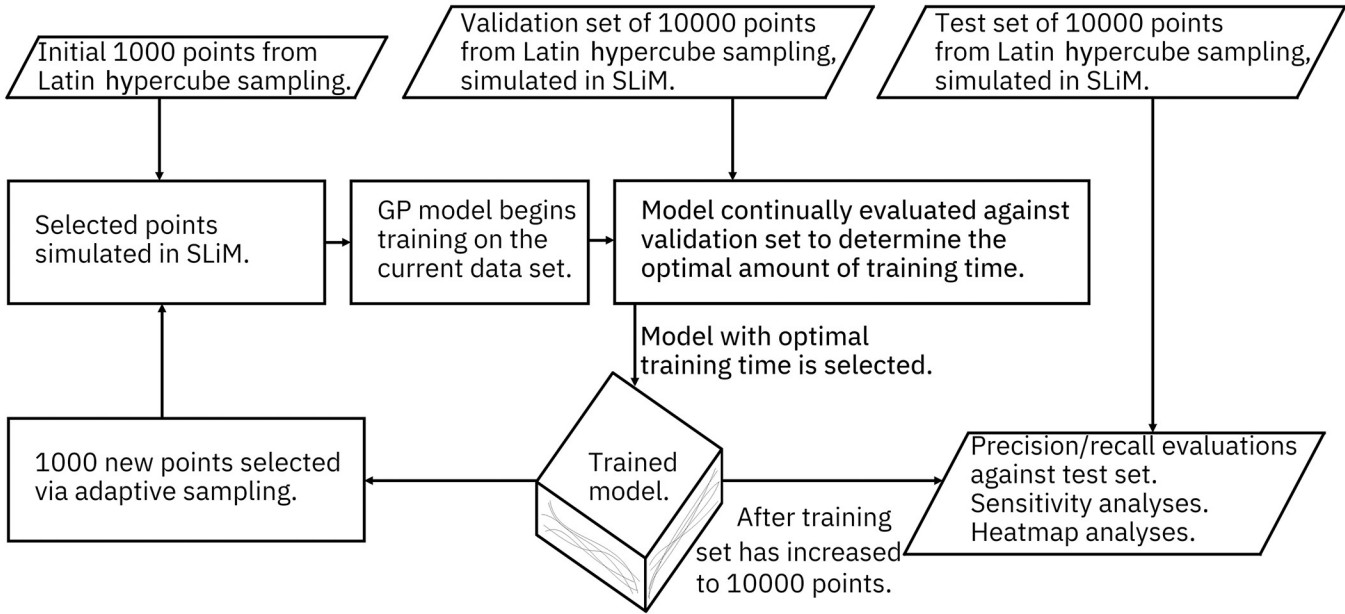

**Fig 1. Gaussian Process model schematic.** The training set started as 1000 randomly sampled points, and the models were considered complete after the training set had been iteratively grown to 10000 points.

1000 iterations. Each saved model was evaluated against the validation data set, and the model that performed best was ultimately selected as the best model for that data set.

Because each prediction made by a GP also comes with a confidence interval, the data set used to train a GP can be adaptively expanded. We implemented two adaptive training strategies and generated a set of ten million Latin-hypercube-sampled parameter points to be evaluated by these strategies for each drive, with the selected points ultimately being simulated by the population model and then added to the training set for the GP model.

The goal of the first sampling strategy was to bolster the data set with additional points where the model has the greatest uncertainty (i.e., the widest 95% confidence intervals). In this strategy, the probability of drawing a given point $i$ with a 95% confidence interval width of $w_i$ is:

$$p_i = \frac{w_i}{\sum w}$$

However, model output uncertainty is not necessarily the same as *outcome* uncertainty. For example, the composite GP might predict a mean of 0.5 and a 95% confidence interval of 0.1 to 0.9, corresponding to a prediction that the drive will suppress, but a high degree of uncertainty about how long it will take to suppress. Conversely, however, the GP is reasonably certain that the final outcome will be population elimination, as the entire 95% confidence interval is greater than 0. We therefore implemented a second adaptive training strategy in order to better develop our understanding of areas of the model with uncertain outcomes. This strategy is built on top of the previous strategy, but linearly scales down the likelihood of choosing points with more extreme predicted outputs. In the composite model, the probability of choosing a given point $i$ with a 95% confidence interval width of $w_i$ and a predicted mean of $v_i$ is:

$$p_i = \frac{w_i * (1 - |v_i|)}{\sum (w * (1 - |v|))}$$

The exact implementation for the suppression rate GP differs, slightly: instead of linearly devaluing points as the predicted value at those points grows more distant from zero, points are linearly devalued as they become more distant from 0.5.

In order to take full advantage of available computing resources, adaptively sampled points were chosen 1000 at a time, after which those points were simulated, and the model was retrained on the new data. This process was repeated until the training data sets for each drive had grown to 10000 points. Of the 9000 adaptively sampled data points, 1000 were sampled using the first of these strategies and 8000 were sampled using the second. Half of the adaptive points were selected using the composite model, and half were selected using the suppression rate model.

## III. Sensitivity analysis

In order to gain an understanding of which input parameters most affect the outcome space of the model, we conducted sensitivity analyses on our GP models using Sobol's method [59]. This method uses a Monte Carlo integration method to decompose variance in the output space into a relative measure of the contribution of each input parameter [59]. The relative contributions of each parameter calculated by this method are termed "sensitivity indices". We calculate sensitivity indices not just for the "first-order" effects of each parameter, but also the "second-order" effects and the "total-order" effects. The first-order effects describe the contribution of varying a parameter to the output when that parameter is varied alone (averaged over variations in all other parameters). The second order indices measure the pairwise synergistic effects of parameter interactions on the output. The total-order effects take into account all first-order, second-order, and higher-order effects of each parameter on the output [83]. These sensitivity analyses were performed using the SALib sensitivity analysis library for Python (SALib version 1.3.8, Python version 3.7.6) [84].

The first step in determining these indices is to generate a Sobol sequence of model input points to test. This is a quasi-random, low-discrepancy sequence, which is designed to minimize gaps in the parameter space [59,85]. These points are arranged in two matrices with $d$ columns and $n$ rows, where $d$ is the number of dimensions of the model (i.e., the number of parameters) and $n$, termed the "base sample," is a freely selected value to which the final accuracy of the analysis is responsive [58,85]. Aside from these matrices, an additional $2*n$ evaluations are required, resulting in $n(2d+2)$ model evaluations in total [85]. Once the required parameter inputs are generated, they are evaluated by the model, after which the sensitivity indices are calculated using the model's output.

The accuracy and reliability of the analysis is generally proportionate to the size of $n$. For example, if $n$ is set to 1000, a sensitivity analysis on a model with 5 parameters would require 12000 queries to the model and would be about as accurate as an analysis of a model with 10 parameters that uses 22000 queries, though the complexity of the parameter interactions can render this relationship non-exact [85]. This type of sensitivity analysis yields confidence intervals for the indices in addition to the indices themselves. Other users of this technique suggest that $n$ should be chosen such that the most important parameters have confidence intervals that are less than 10% of the magnitude of the indices themselves [86]. We used an $n$ of one million for our sensitivity analyses, providing very tight confidence intervals.

Nothing about the underlying population model precludes a direct sensitivity analysis other than computation time. However, a sensitivity analysis for one drive with $n$ of one thousand would require almost as many simulations as the entire training and validation of a GP model for that drive, and such a sensitivity analysis would likely be fairly inaccurate, with very wide confidence intervals.

## Results

### I. Population model

**Default population model.**   Our population model simulates the release of one of three different types of suppression gene drives into an island population of rodents to assess if the drive can eliminate the population. We first examined the population model in the absence of any gene drive release in order to verify that the model dynamics are reasonable and generally match our expectations of rat population dynamics.

The default parameters of the model call for an island side length of two kilometers and a density of 1000 individuals per square kilometer (see Table 1), resulting in an estimated carrying capacity of 4000 individuals. Actual density within the population is a function of a density tuning coefficient that is calculated at the outset of the simulation. Due to minor discrepancies between the function that calculates this value and the model itself (e.g., individuals in the model are not uniformly distributed, along with other stochastic effects), the actual capacity of the system is an emergent property that varies slightly from the estimated value. With default parameters, the capacity of the system was about 6% higher, with the population tending to fluctuate between 4100 and 4400, with an average size of about 4250 (Fig 2, left panel).

The age distribution of individuals in the model is dependent on a flat survival rate parameter, as well as on density dependent competition faced by migrants and newborns. On average, with default parameters, 90% of the population was 12 time steps old or less, while about 1% of the population consisted of individuals who had survived 24 time steps or more (Fig 2, center panel). This matches our expectations of a high death rate among young rats and increasingly few individuals surviving to older ages [61,87]. The average age in the population was 4.6 time steps, while the life expectancy of individuals who survived to adulthood (i.e., individuals who survived their first time step) was 5.6 time steps, equivalent to around 9 to 14 months and 11 to 17 months respectively assuming a time step duration of two to three months. This is close to the general dynamics we expect in a real population.

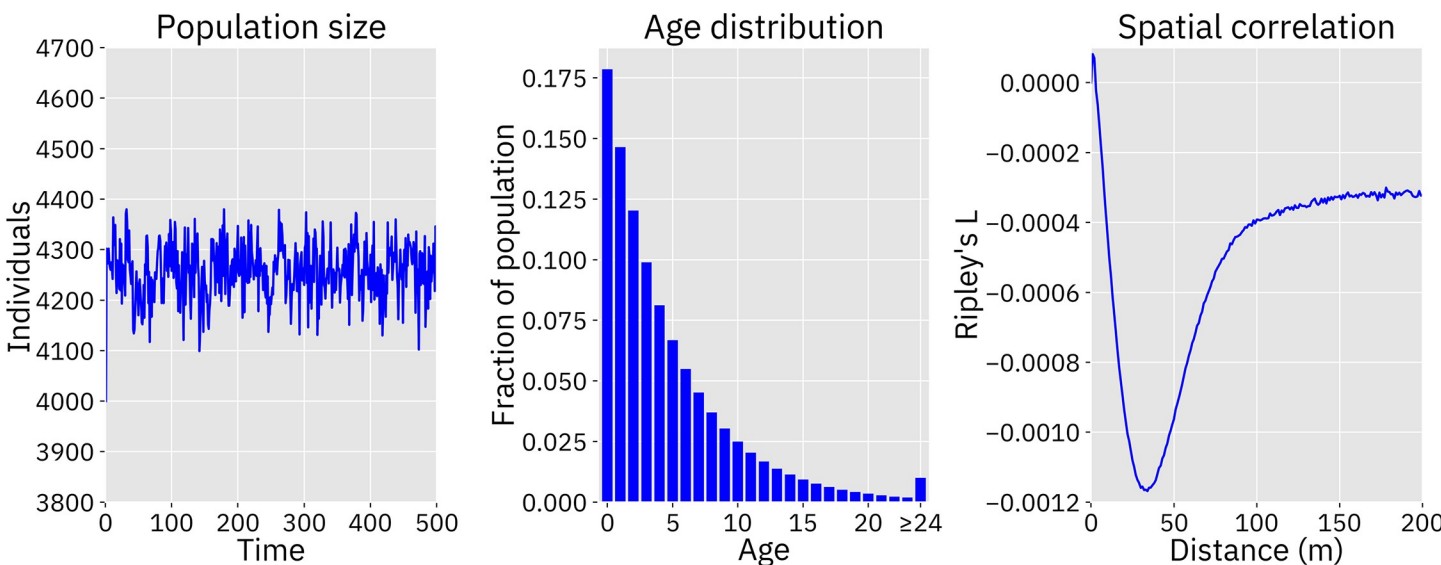

**Fig 2. Population dynamics without gene drive release.** Left panel: Population size fluctuations over time are shown after the population is initialized at the approximated capacity of 4000. The average population size was 4257 individuals. Center panel: The average frequency of each age, as determined by tabulating the ages of each individual between time step 100 and 500 of the simulation. Right panel: Ripley's L at length scales from 0 to 200 meters. This plot shows the difference between Ripley's L statistic in the simulation and the expected value of a random distribution. Negative values indicate that a population is more dispersed than a randomly distributed population. Density dependent competition takes place at a maximum distance of 75 meters and is more intense between close-together individuals, resulting in a slightly dispersed population.

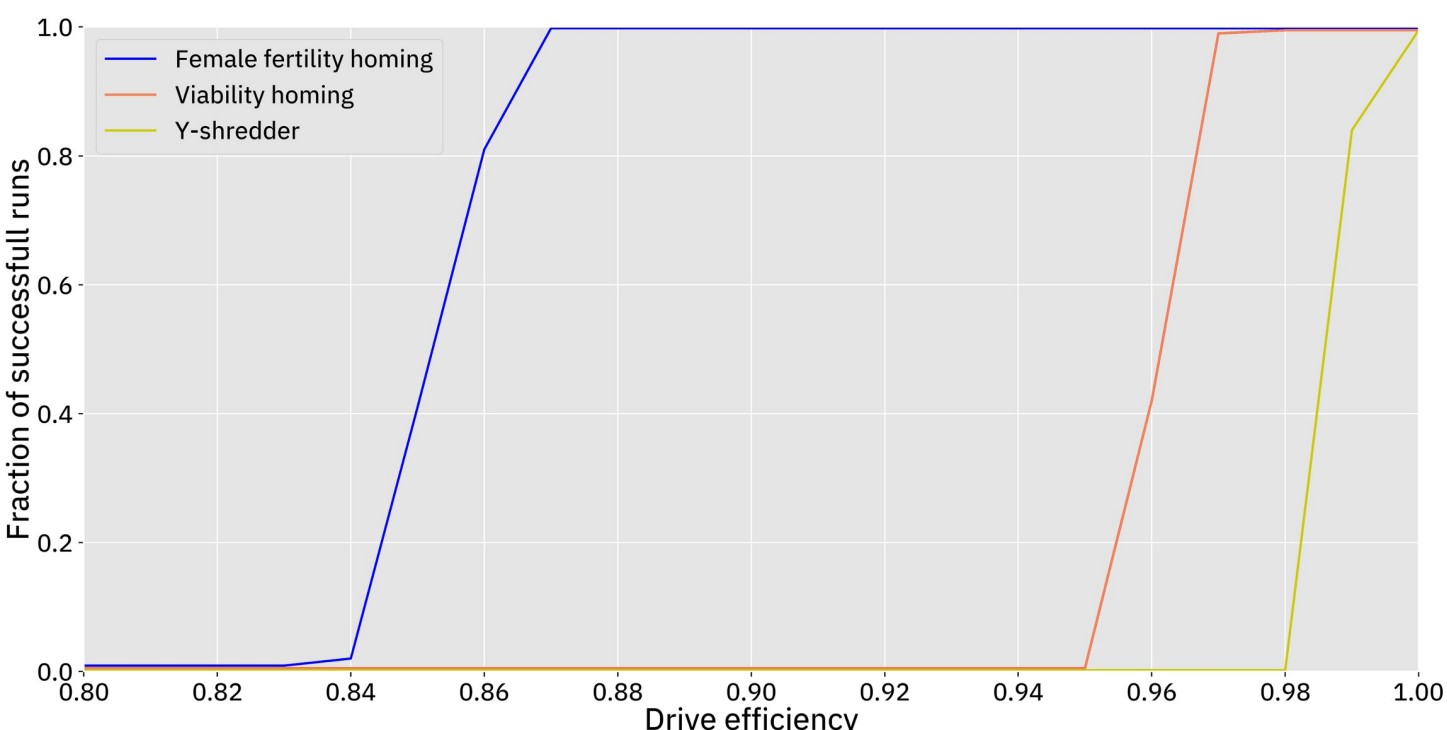

**Fig 3. Drive efficiency required to eliminate the population using the default model.** Drive efficiency was varied in 1% increments, leaving other parameters at default values, with 100 simulations conducted at each point. A "successful run" is defined as a simulation in which the population is completely eliminated within 500 time steps.

To measure deviations from spatial homogeneity between individuals in the population, we estimated Ripley's L function within our model. This function describes whether a set of points are dispersed or clustered at any given length scale as compared to randomly distributed points [88–90]. We assessed the population after 20 time steps and found that individuals tended to be somewhat more dispersed than random at length scales below 100 meters. Beyond this distance, the population is only marginally more dispersed than a random set of points. This correlates fairly well with the default maximum competition distance parameter of 75 meters, indicating that density dependent competition appears to have discouraged clustering as intended (Fig 2, right panel).

**Gene drive performance in the default population model.**   After verifying the demographic characteristics of the model, we tested the ability of each of the three gene drives to suppress the population. We found that each of the three drives is capable of suppressing the population given a sufficiently high drive efficiency, no resistance, and otherwise using default parameters. The transition between invariable failure and invariable success occurs rather abruptly as efficiency is increased beyond a threshold (Fig 3).

We first assessed the homing drive with a haplosufficient female fertility target. This drive required the lowest efficiency in order to achieve success. At 84% efficiency, 2% of simulations resulted in complete suppression, increasing to 100% of simulations at 87% efficiency. We also found that efficiency impacted the average time it took to eradicate the population. With 85% efficiency, the drive took an average of 377 time steps to eradicate the population when it succeeded. With 95% efficiency, the drive took an average of only 158 time steps (Figs 3 and 4). With 85% efficiency, suppression usually occurred only after a moderate amount of fluctuation in the population size, while the drive with a 95% efficiency caused the population to decline

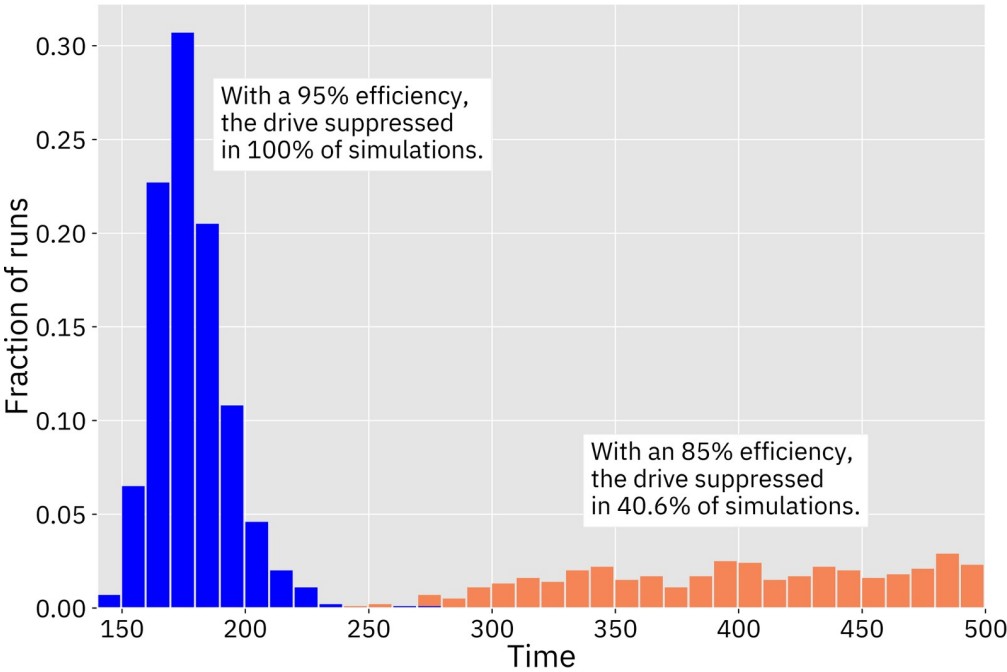

**Fig 4. Time from release until suppression for the drive targeting female fertility.** Number of time steps until population eradication for the homing drive with a female fertility target using default parameters and a drive efficiency of 95% (blue) and 85% (orange). One thousand simulations were performed for each drive.

smoothly once the drive had spread (Fig 5). When drive efficiency was less than 84%, the population was reduced to a smaller equilibrium size, though there also appeared to be shifting migration patterns that ambiguously resemble the chasing phenomenon that was more clearly seen in a discrete generation model [37].

The homing drive with a haplosufficient viability target required a higher efficiency to succeed. This drive was able to eliminate the population when it had an efficiency of above 95% (Fig 3). With a 96% efficiency, the drive took an average of 340 time steps to completely suppress the population and exhibited fairly similar dynamics to the drive with a female fertility target with an 85% efficiency. Insufficient drive efficiency resulted in the same type of reduced equilibrium population as observed in the drive with a female fertility target. This reduced efficiency compared to the female fertility drive is due to the lower genetic load from the drive at equilibrium, since drive alleles are removed in both male and female individuals.

The Y-shredder required the highest efficiency to succeed. The drive was only successful when it had 99% or 100% efficiency (Fig 3), in which case it completely suppressed in 84% and 100% of simulations respectively. Notably, "efficiency" in the Y-shredder is achieved by destruction of the Y-chromosome in germline cells, a fundamentally different process than the homology-directed repair required for the copying of homing drives. Thus, efficiency levels cannot be directly compared between these different types of drives, and it is unclear based on previous studies how difficult it may be to engineer each type of drive with a requisite efficiency. With a 99% efficiency, the Y-shredder took an average of 380 time steps to eradicate the population. When this drive had too little efficiency, the population size actually increased. This is because a female-biased sex ratio of the population resulted in more total offspring in each time step (the number of females that each male can reproduce with is not limited in our model), which was also found by Prowse *et al.*[50]. This limitation may partially account for

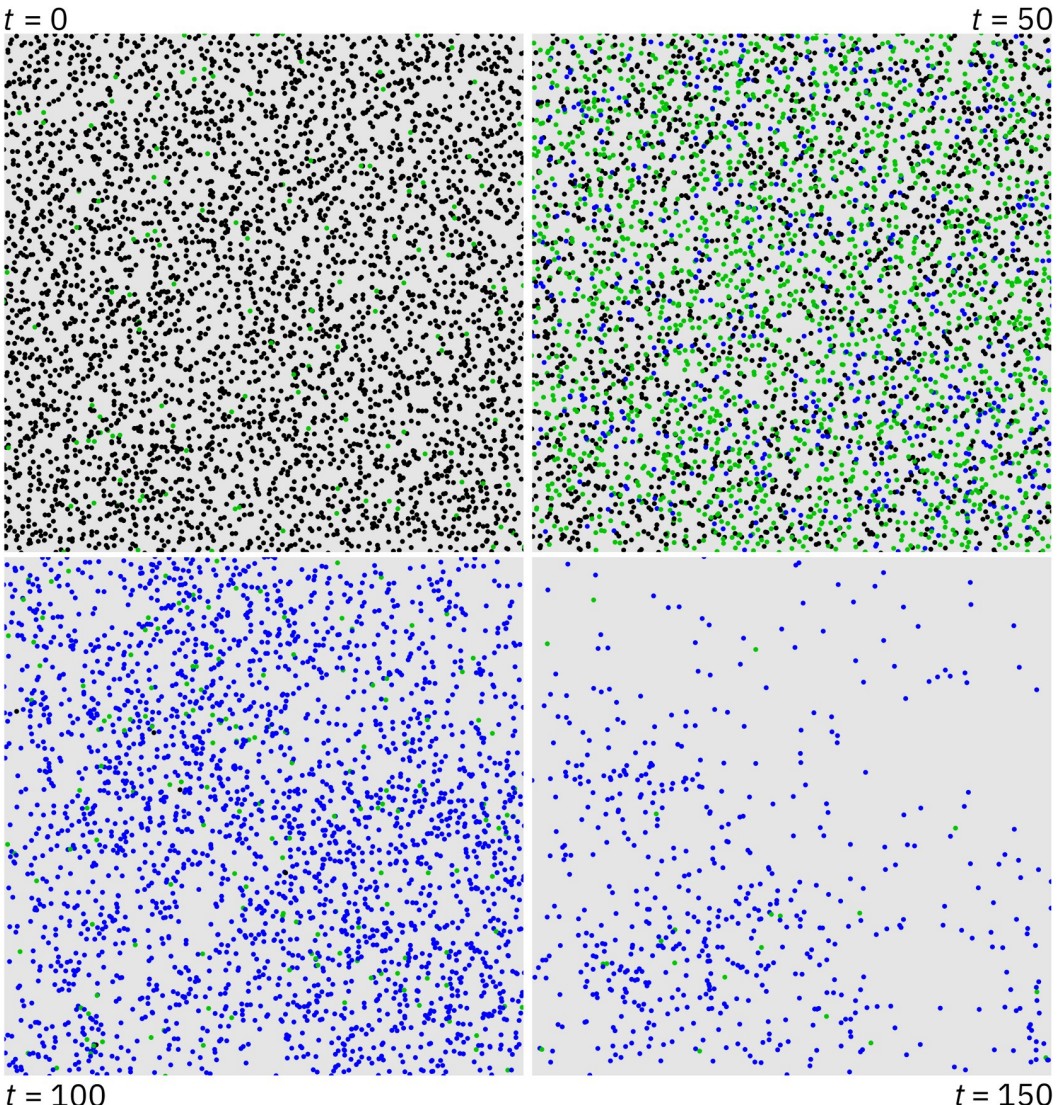

**Fig 5. Gene drive spread over time for the drive targeting female fertility.** A series of snapshots showing the progress of a single simulation of the homing drive with a female fertility target using default parameters and a drive efficiency of 95%. Black dots are wild type, green dots are drive/wild-type heterozygotes, and blue dots are drive homozygotes. The upper left panel depicts the time step at which the drive was introduced to the population. At first, the drive rapidly spreads through the population (upper right). After the drive has reached a sufficient frequency, the population dwindles in size (lower left) until almost all possible pairings only produce sterile offspring (lower right), leading to population elimination shortly thereafter.

the Y-shredder's worse performance than in that study, which used panmictic populations and an absolute limitation on the number of females each male could mate with during each time step.

## II. Gaussian process framework

We created two different types of Gaussian process (GP) meta-models, differentiated by the outputs from the underlying population model used to train them. Both models are designed to predict whether the drive will eliminate the population within 500 time steps of the drive

release. The output used to train one model was a composite value that comprises time to suppression if suppression occurs and final population size if suppression does not occur. The output used to train the other model was simply the portion of simulations in which suppression occurred. We gauge the success of our GP models by two factors: first, how much faster the meta-model evaluates data points than the underlying population model, and second, how accurately the meta-model predicts the outcomes of the underlying population model.

**GP evaluation speed and training time.** We found that both GP models are several orders of magnitude faster than the underlying model. The runtime of the underlying model is highly dependent on the parameters selected. A simulation of a rat population over 500 time steps with default parameters and without a gene drive takes approximately 30 seconds. A simulation with fewer individuals and a highly effective gene drive can finish in just a few seconds. However, simulations with larger numbers of individuals (e.g., for much larger islands or denser populations) can take longer than one hour. These runtimes are representative of speeds attained on Cornell's BSCB computing cluster using Xeon E5 4620, Xeon E7 4830, and Xeon E7 4850 CPUs. Further, given the stochastic nature of the model, numerous runs per data point are necessary to enable reliable conclusions to be drawn.

The GP model, on the other hand, once trained, evaluates all data points with equal speed, at a rate of approximately 45,000 points per second. Producing a 1000 by 1000 point heatmap analysis takes approximately 23 seconds. Sensitivity analyses requiring 24 million (drives without resistance) or 28 million (drives with resistance) model evaluations took 45minutes and one hour, respectively (the majority of that time is spent generating the data points to be evaluated). These runtimes are representative of speeds attained on a desktop computer using an i9-9900K CPU and a GeForce 2080Ti GPU.

The time spent training the models in the first place was bottlenecked mostly by the evaluation speed of the population model. The process consisted of several iterations of running 1000 simulations, training the GP model, and then choosing 1000 new points to simulate (see methods). Each point in the parameter space used to train the GP model consisted of an average of twenty simulations for that point. Most points took an hour or less to simulate, but the longest points took about 24 hours. Thus, even when sufficient CPU resources were available to run 1000 simulations in parallel, this portion of the training process always took a full 24 hours. The training time required by the GP model increased as the size of the training set increased. With 1000 points, the time to train was about 20 minutes. This scaled up to about 3 hours as the data set reached 10000 points. The selection of new points was done by evaluating a set of ten million points using our adaptive sampling algorithm, which took only a few minutes. There was no noticeable difference in the required training time for the suppression rate model and the composite model, nor is there a noticeable difference in evaluation time.

**GP accuracy measurements.** We assessed the accuracy of our GP models in terms of root-mean-square error (RMSE), precision, and recall (see methods). Each of the models was assessed against a test set consisting of semi-random points from a Latin hypercube sampling of the parameter space. Models trained on drives including resistance were also assessed against the test sets for the versions of those drives without resistance. In most cases, the final GP models appear to be excellent predictors of the underlying population model (Table 3).

As adaptively sampled data points were added to the initial 1000 training points, the model quality increased rapidly at first, but with diminishing returns. Generally, the first and second set of adaptively selected points increased model performance substantially. The quality measurements for the GP model occasionally declined slightly after some batches of adaptive sampling, though usually these were minor fluctuations. Generally, model quality continued to improve throughout the course of adaptive sampling, indicating that further improvements may be possible with larger data sets (Figs 6 and S1).

**Table 3. Gaussian Process Model Accuracy.**

| GP Model | Rate of suppression in test set | RMSE (composite) | Precision (composite) | Recall (composite) | RMSE (sup. rate) | Precision (sup. rate) | Recall(sup. rate) |
|---|---|---|---|---|---|---|---|
| F. fertility homing w/o resistance | 0.21 | 0.09 | 0.94 | 0.95 | 0.07 | 0.97 | 0.95 |
| F. fertility homing with resistance | 0.16 | 0.13 | 0.78 | 0.90 | 0.05 | 0.96 | 0.84 |
| Viability homing w/o resistance | 0.03 | 0.06 | 0.88 | 0.96 | 0.02 | 0.98 | 0.89 |
| Viability homing with resistance | 0.02 | 0.08 | 0.87 | 0.67 | 0.01 | 0.99 | 0.85 |
| Y-shredder | 0.05 | 0.13 | 0.94 | 0.69 | 0.03 | 0.93 | 0.84 |

See S1 Table for additional comparisons.

Model performance was better for drives with larger areas of success, such as the homing drive with a female fertility target. These differences were especially large before adaptive sampling. The models for drives with very small areas of success had trouble detecting those areas, and thus suffered from higher rates of false negatives.

The model trained on the composite output from the population model generally had more false positives, while the model trained on the suppression rate from the population model had more false negatives. The suppression rate models tended to outperform the composite models more when the models included drive resistance (possibly because the minimum population level could vary greatly depending on when and where r1 resistance alleles formed, resulting in a more stochastic composite value). However, the composite models tended to have much

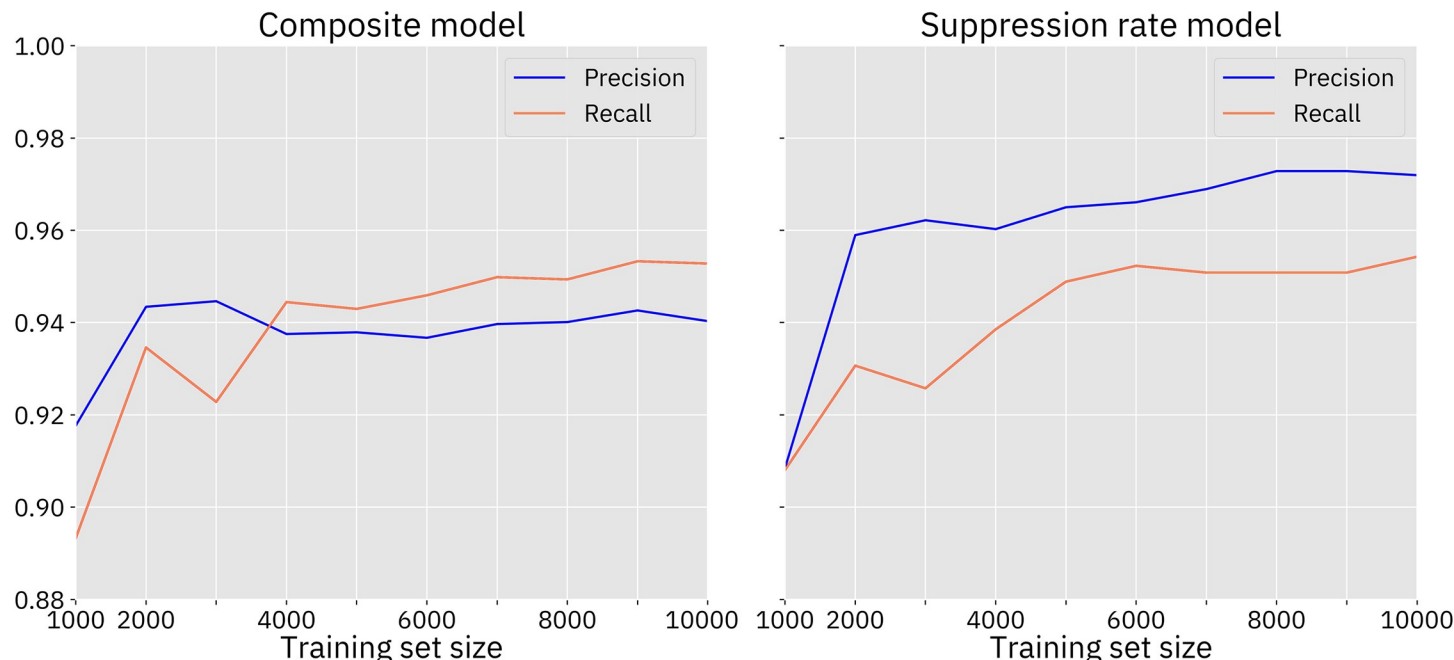

**Fig 6. Model quality, female fertility homing drive without resistance.** The model was evaluated against the 10000-point Latin Hypercube sampled test set. This drive has a larger area of success than all of the other drives, and thus, the GP model performs relatively well, even before adaptive sampling, compared to the other models. The first few adaptively sampled data sets substantially improved the performance of the model, after which improvements were smaller. See S1 Fig for similar plots for the other models.

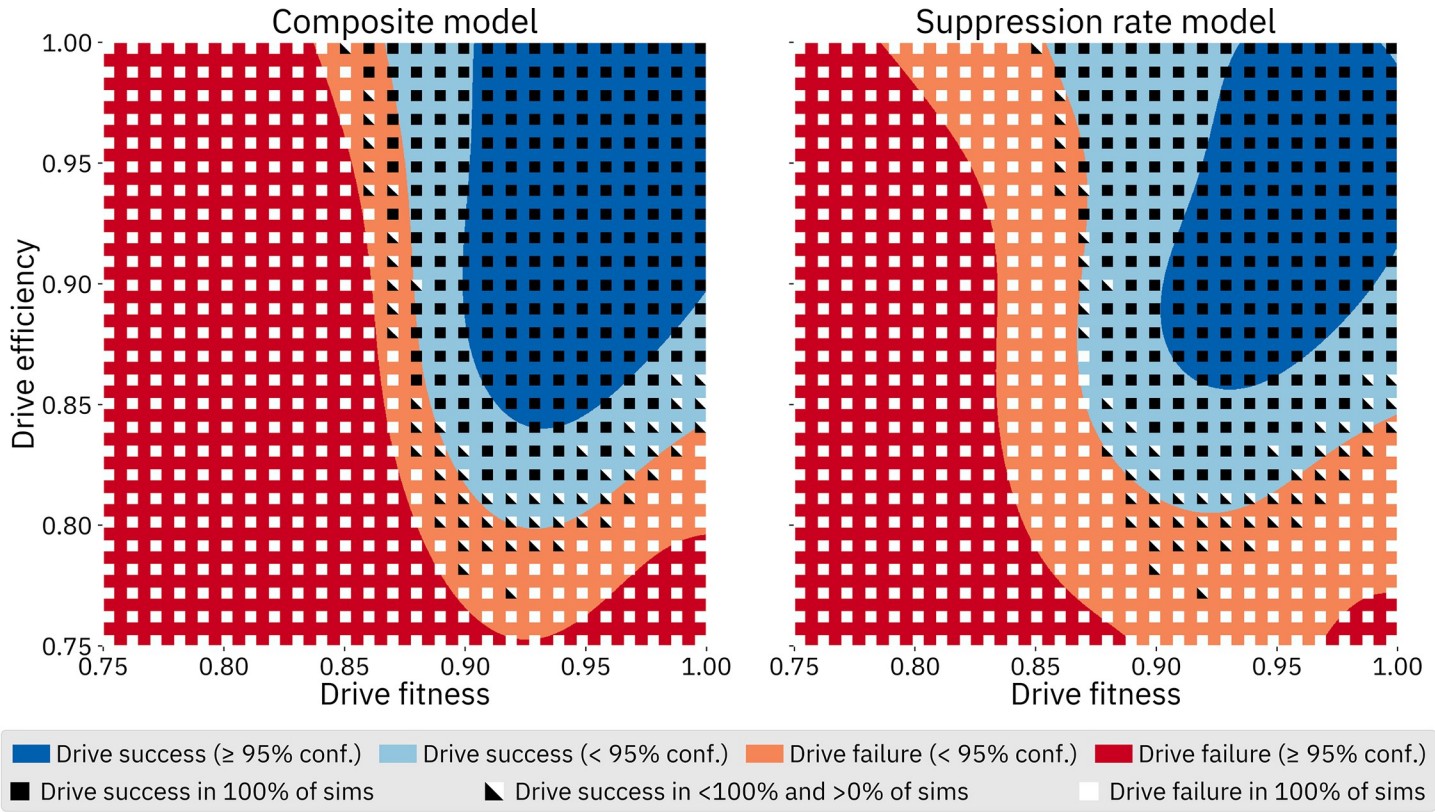

**Fig 7. Model comparison, female fertility homing drive without resistance.** For this comparison, drive fitness and efficiency were varied while other parameters were kept at default values. A 1000 by 1000 array of queries was made to both GP models; color indicates model predictions of drive success or failure, as well as confidence. Black and white square dots show results from simulations using the underlying model. Each square dot denotes the result of twenty simulations performed at each point. This overlay serves as a visual validation of the GP framework; the actual model behavior in this parameter range (i.e., how drive success depends on model parameters) is discussed in more detail in the "Selected Model Outcomes" section below. See S2–S5 Figs for similar plots for the other models.

narrower confidence intervals, and is more confident about its correct predictions. As a visual validation of the two models, Figs 7 and S2–S5 show the overlay of a plot from each model with actual results of the underlying population model.

It is conceivable that there are qualitative differences between the two model types that reflect different ways in which the models learn, yet are not easily described by the statistical measures calculated in Table 3. For example, based on visual observation, it appears that the composite model predicts boundaries between success and failure that are more homomorphic to the underlying model than the suppression rate model. However, the boundaries predicted by the composite model are often translated somewhat from where the drive transitions from success to failure in the underlying model. On the other hand, the suppression rate model appears to better align with the transition from success to failure in the underlying model, but the predicted boundary is less similar in shape. In other words, if a line between success and failure in the underlying model could be described by $f(x) = ax^2+bx+c$, the composite model appears to perform better at finding $a$ and $b$, while the suppression model appears to perform better at finding $c$. This makes intuitive sense: the suppression rate model is being trained on exactly what we are asking it to predict, so it is lining up with the underlying model very well. The composite model, on the other hand, is trained on a function that allows the model to learn more about drive performance, but it is not actually being trained on what it is being

asked to predict. However, model characteristics of this nature are difficult to precisely measure, and we did not assess these differences quantitatively.

Overall, the models for the homing drives without resistance appear to be reasonably accurate. The models for the homing drives with resistance as well as for the Y-shredder are somewhat less accurate, apparently due to the relatively smaller area of the parameter space in which these drives are successful. Yet, even these models are of sufficient quality that sensitivity analyses performed on them should be reasonably reflective of the underlying population model.

### III. Sensitivity analyses

We conducted variance-based Sobol sensitivity analyses on each of our GP models. The first set of analyses is based on samples drawn from across the entire parameter space for each drive, with the exception that drive fitness and efficiency for the homing drives without resistance were bounded to the range [0.75, 1] since this allows for direct comparison to the models of those drives with resistance in which those parameters are restricted to that range. The resulting sensitivity indices of the model parameters are shown in Fig 8. The second order sensitivity indices for this analysis are shown in S6–S10 Figs.

These sensitivity analyses confirm our presuppositions that drive fitness and efficiency are key parameters for each of the three drives. Resistance also has a substantial effect on the model. In the female fertility homing drive modeled with resistance, the resistance parameter dominates all other parameters. Interestingly, the simulation of resistance also has a large impact on the model for the viability homing drive. However, the parameter itself does not appear to contribute much to the variance of the outcome. Instead, the relative importance of the drive efficiency parameter appears to have increased, with the importance of all other parameters reduced (see the Discussion for an analysis of this result).

It appears that the different types of drives respond differently to demographic parameters as well. The homing drives were more responsive to survival rate than any other demographic parameter. Yet, this parameter seems to be of low importance for the Y-shredder. Interaction distance was found to be an important parameter for the Y-shredder but had very little effect on the homing drives. Average dispersal was also a somewhat important parameter in the Y-shredder analysis, while having only small effects in the other models. Most of the models were at least somewhat responsive to litter size. Migrant frequency has the largest effect on the female fertility homing drive. Island area affects the female fertility homing drive substantially when resistance is simulated, likely due to the larger number of individuals in which resistance alleles could be generated.

The zero-results also provide insight into the system. None of these drives are considered to be frequency-dependent, and the models agree: across the board, release percentage has little to no effect, despite the fact that this parameter ranged from only 1% of the population up to 50% of the population.

The adult dispersal speed multiplier was also found to have essentially no impact on the outcome. In our population model, individuals who are selected as migrants first move to a new location and then experience density dependent mortality. From these sensitivity analyses, we can therefore surmise that it is the latter of these two processes that usually has the greater impact on the model. Each time step, both migrants and newborns are dispersed according to the average dispersal parameter. The migrant frequency ranges from 0% to 50%, while the litter size varies from 2 to 8, meaning an average of 100% to 400% of the population size. Thus, at any given moment in the simulation, the vast majority of migration in the system can be attributed to newborns, leaving the fact that migrants can move further distances less important.

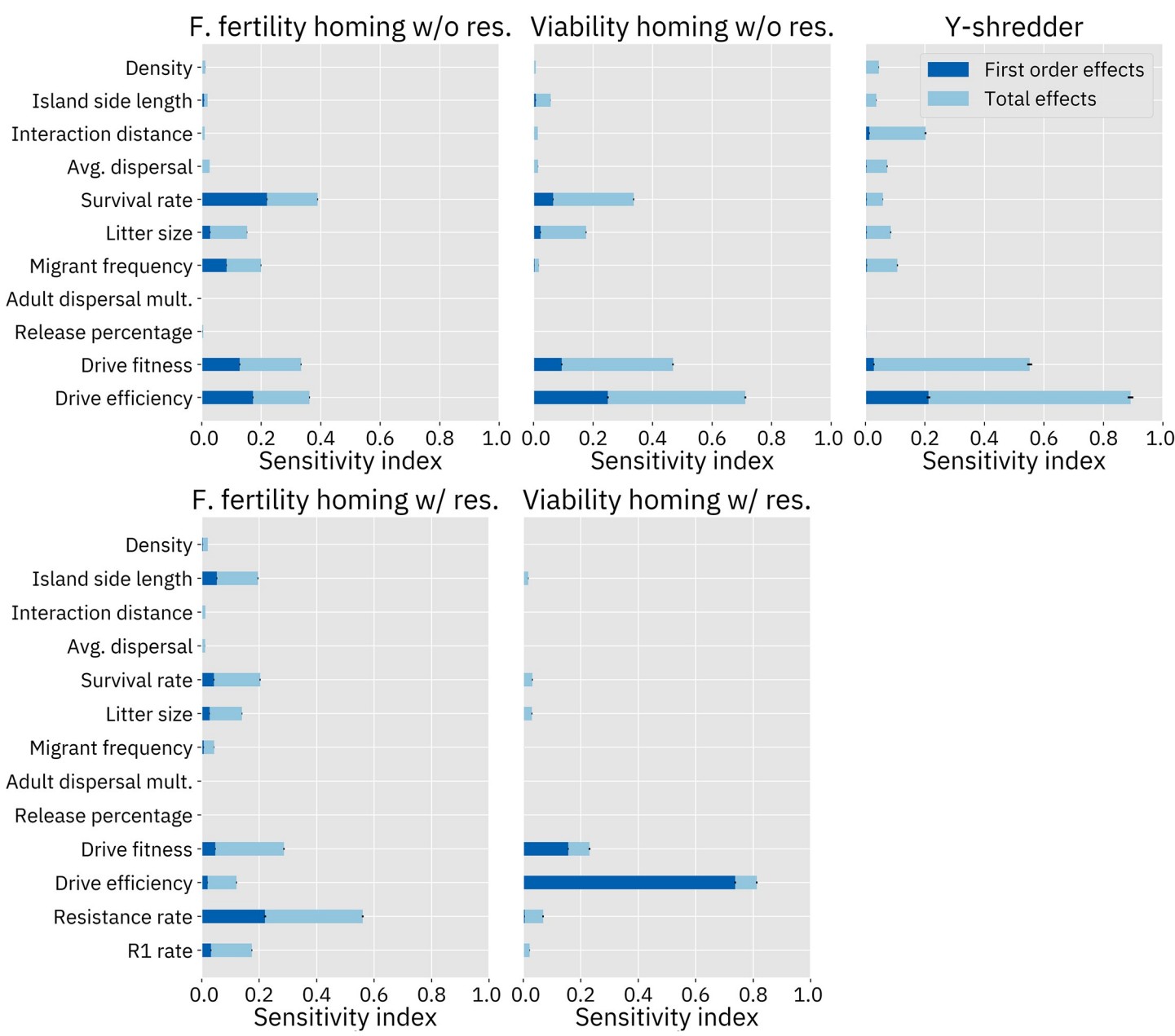

**Fig 8. Sobol sensitivity analysis.** "First order effects" describe the effects of varying a single parameter. "Total effects" include the first order effects of the parameter, as well as potential synergistic interactions between that parameter and one or more other parameters. Analyses of drive models without resistance sampled 24 million points from the parameter space. Analyses on drive models with resistance sampled 28 million points. These analyses are from the models trained on the suppression rate due to their somewhat better accuracy. Analyses of the composite model were similar.

Given the critical importance of drive efficiency and fitness, we anticipate that any real-world release will be predicated on the successful construction of a drive with relatively high efficiency and low fitness costs. If this can be achieved, it will become important to determine which other parameters become the most important factors in determining the ultimate outcome of the release, as these parameters may be of only middling importance elsewhere in the parameter space. To study this question, we performed a second set of sensitivity analyses in a subspace of the full parameter space where drive fitness and efficiency were set to be equal and

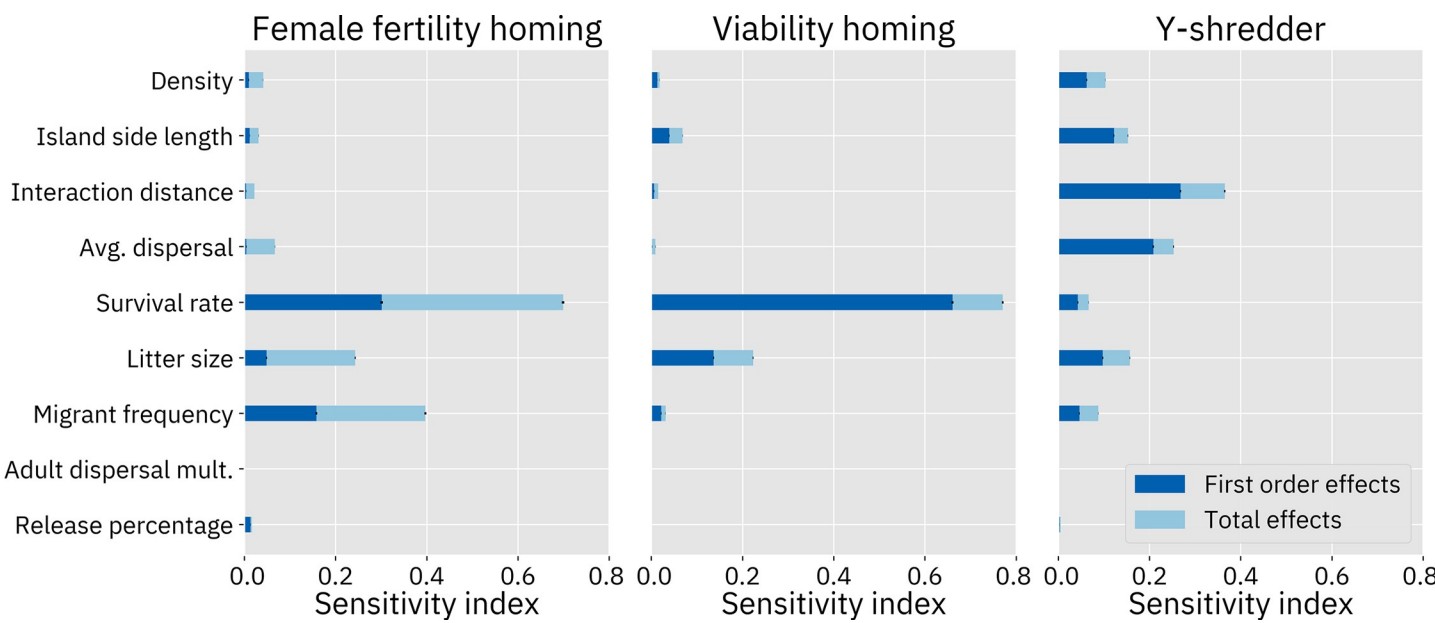

**Fig 9. Sobol sensitivity analysis with fixed fitness and efficiency.** For the female fertility homing drive, fitness and efficiency were both fixed at 90%. For the viability homing drive, fitness and efficiency were both fixed at 97%. For the Y-shredder, fitness and efficiency were both fixed at 99%. The models without resistance were used. "First order effects" describe the effects of varying a single parameter. "Total effects" include the first order effects of the parameter, as well as potential synergistic interactions between that parameter and one or more other parameters.

fixed at 99% for the Y-shredder, 97% for the viability targeting homing drive, and 90% for the female fertility targeting homing drive (Fig 9). These levels were chosen because they resulted in eradication success in the majority of cases when other parameters are at default levels, and are likely around the minimum that a "release candidate" would need to be able to completely suppress a population without other control measures, though not with much leeway for reducing efficiency any further. For purposes of this set of analyses, models without resistance were used, due to the superior accuracy of those models.

The most important parameters in these analyses tend to be the same ones that were important across the entire parameter space (with the obvious absence of the fixed parameters). However, the sensitivity indices of the remaining parameters are higher than in the analysis of the entire parameter space because these parameters now drive the entire variation of the output. In the sensitivity analysis of the Y-shredder with fixed fitness and efficiency, the first-order effect indices now account for a much greater proportion of the overall variation. This is because most of the impact caused by spatial parameters (such as interaction distance, average dispersal, island size, and population density) in the sensitivity analysis of the full parameter space are described by second order synergistic effects between those parameters and drive efficiency (see S10 Fig).

## IV. Selected model outcomes

While sensitivity analysis is a powerful tool for understanding the relative importance of the input parameters, it does not actually offer an understanding of the nature of the interaction between any given parameter and the output space. We therefore also performed a set of more traditional analyses wherein we varied two or three parameters at a time, with the parameters we chose to vary being informed by the results of our sensitivity analyses. The composite GP models were used for these analyses since those models tend to have higher confidence in their

correct predictions. Unless assessing a resistance parameter, models trained without resistance were used, due to the higher accuracy of those models.

As expected, drive fitness and efficiency are key parameters for all of the drives. An analysis wherein fitness and efficiency were varied while other parameters were kept at default values shows that if these two drive characteristics are not sufficiently high, the drive cannot succeed (Fig 10). This analysis also highlights the substantially larger area of success for the homing drive with a female fertility target as compared to the homing drive with a viability target, which in turn has a substantially larger area of success than the Y-shredder.

Notably, it appears that if the homing drive with a female fertility target has a low efficiency, the drive actually performs better if it has a moderate fitness cost than if it has either a high or low fitness cost (this phenomenon was present in the underlying population model, see Fig 7). The root cause of this unexpected result is that in our population model, drive fitness is implemented as a survival rate multiplier that is applied every time step and which is equal for both drive homozygotes and drive heterozygotes. Even when the drive has less than 100% efficiency, the drive is still able to quickly propagate through the population. However, after the population consists entirely of drive carriers (i.e., individuals that have at least one copy of the drive), there are still many possible pairings that yield fertile offspring. In the absence of a drive fitness cost, this results in the population declining to a new, lower equilibrium size. When the drive has a fitness cost, this cost acts to remove wild-type alleles from the population in addition to drive alleles, providing additional suppressive power. Notably, this phenomenon is substantially decreased for drives with a higher rate of resistance formation.

Litter size and migrant frequency also appeared to be important parameters for all of the drives, especially in the sensitivity analyses wherein drive fitness and efficiency were fixed. We thus assessed the effects of varying these two parameters while similarly fixing drive fitness and efficiency (Fig 11). We found that the drive performs better in populations with a higher frequency of migrants. Smaller average litters enabled greater drive success in the homing drive with a viability target and the Y-shredder but did not substantially affect the homing drive with a female fertility target with these parameters. Litter size has a larger effect on the outcome of the female fertility homing drive in other areas of the parameter space (S11 Fig).

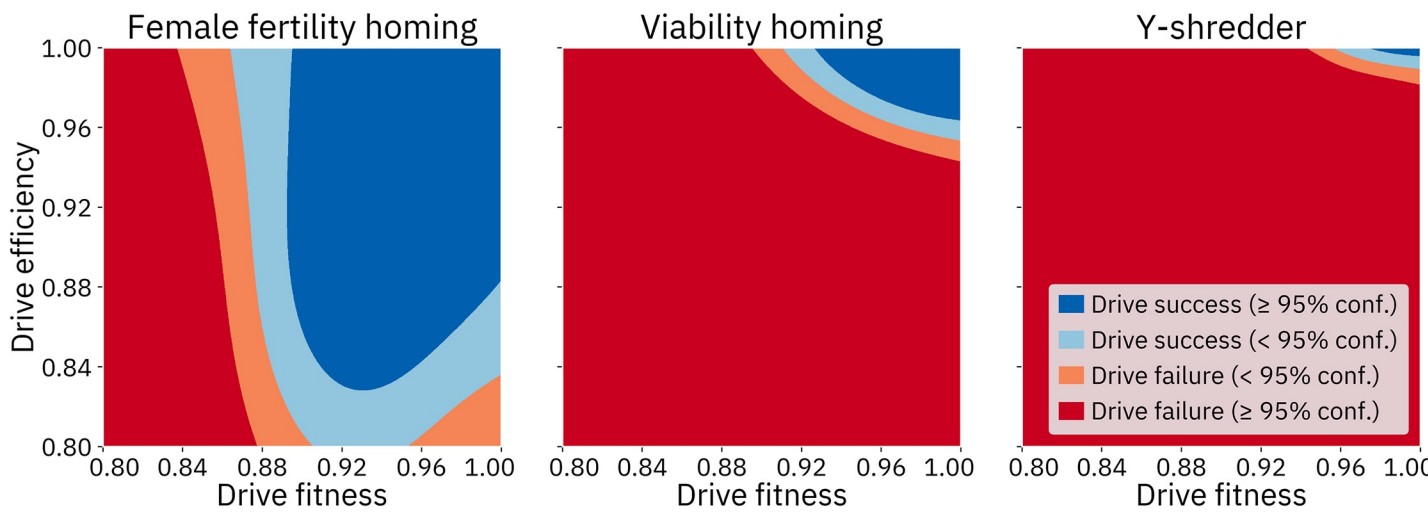

**Fig 10. Comparison of the three drives with varying fitness and efficiency.** Other parameters were fixed at default values. For each panel, a 1000 by 1000 array of queries was made to the GP model for that drive.

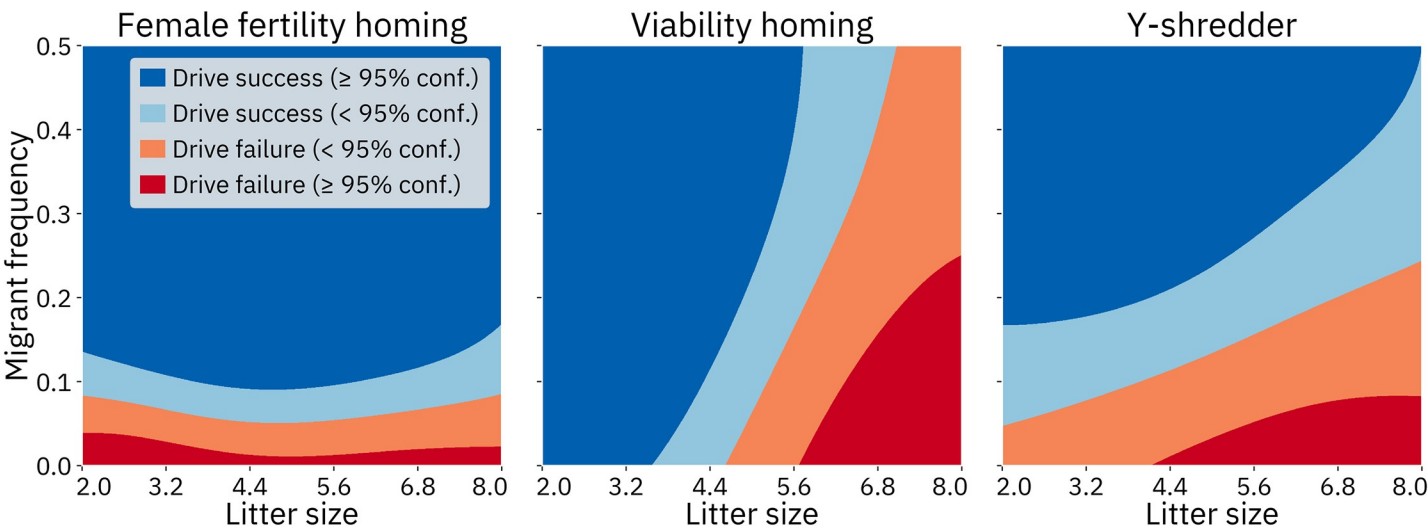

**Fig 11. Comparison of the three drives with varying litter size and migrant frequency.** For the female fertility homing drive, fitness and efficiency were both fixed at 90%. For the viability homing drive, fitness and efficiency were both fixed at 97%. For the Y-shredder, fitness and efficiency were both fixed at 99%. The models without resistance were used.

In the two homing drives, the third parameter with a high sensitivity index is survival rate. In order to understand the effects of this parameter, we analyzed the drive fitness and efficiency required for a successful drive while varying the survival rate (Figs 12 and S12). This analysis suggests that a population with a higher mortality rate could be suppressed by a gene drive with a much lower fitness and efficiency. The homing drive with a viability target has a substantially smaller area of success.

In the Y-shredder model, interaction distance and average dispersal distance stand out as important parameters, in contrast to the other models. We varied these two parameters with drive fitness and efficiency fixed at 0.99, matching the sensitivity analysis with those parameters fixed (Fig 13). As expected, shorter interaction distances enable the drive to succeed, likely

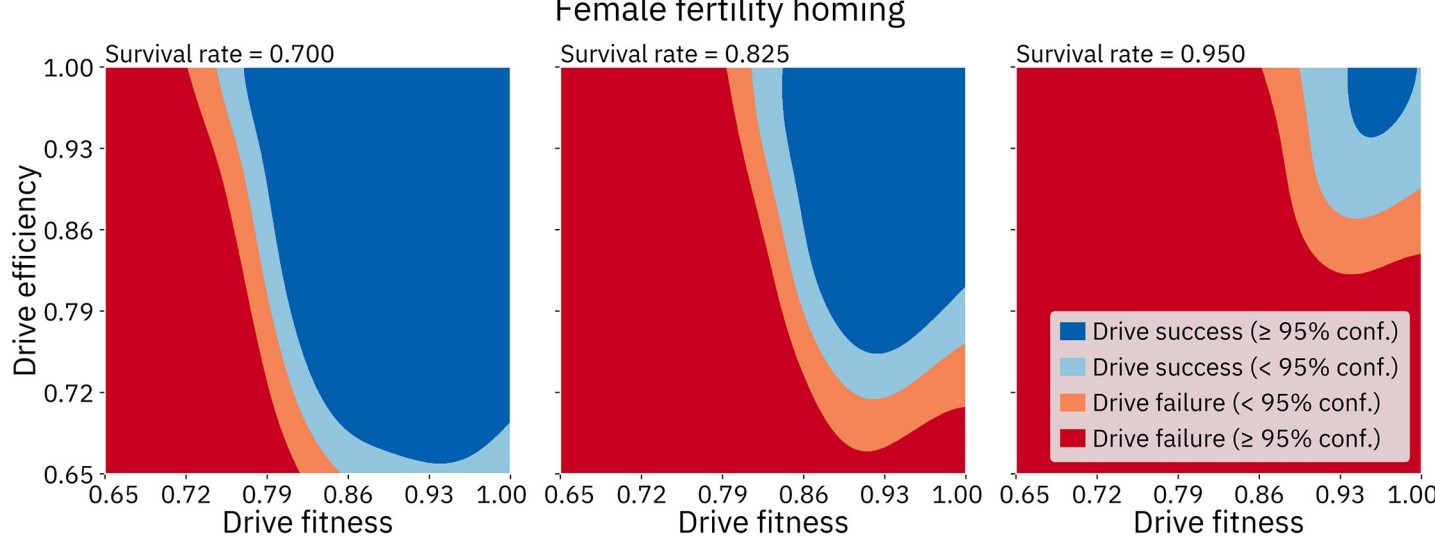

**Fig 12. Female fertility homing drive with varying fitness, efficiency, and survival rate.** Other parameters were fixed at default values.

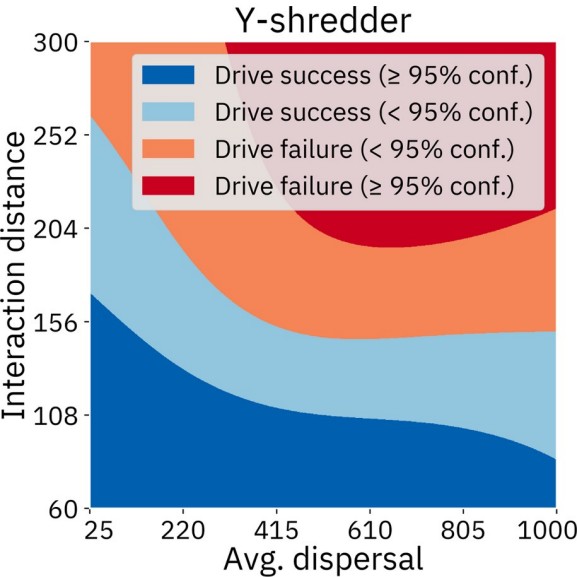

**Fig 13. Y-shredder with varying interaction distance and average dispersal.** Drive fitness and efficiency were both fixed at 99%. Other parameters were fixed at default values.

because females find it increasingly difficult to find mates once the drive has reached a high frequency. A lower average dispersal distance also helps the drive succeed. When individuals in the population tend to move around farther, areas where the drive is locally succeeding can potentially be disrupted by males migrating from areas where the drive is not present at as high a frequency, resulting in the drive failing to suppress the population.

Resistance, when it was modeled, also had a large impact on the models for the homing drives. We compared the performance of the two homing drives with and without resistance and found that the area of drive success of both drives shrinks rapidly in response to resistance (Fig 14).

Any number of two-factor-at-a-time analyses can be conducted using these meta-models. The code to generate heatmaps for any other parameter pairs, as well as animated heatmaps (wherein a third factor is also varied) using fully pre-trained models is available online at https://github.com/MesserLab/GeneDriveForSuppressionOfInvasiveRodents.

## Discussion

### Population model

In this study, we developed a simulation model of a spatially continuous rat population to examine the ability of different types of suppression gene drives to eliminate such a population. Our model incorporates several features intended to increase the ecological realism of the simulated rat population, and which could affect the dynamics of gene drive spread in important ways. These features include local competition and mate choice, migration of offspring and migrant adults, and overlapping generations. By training a GP as a meta-model for our simulation model, we were able to conduct a comprehensive sensitivity analysis to assess the relative importance of the various model parameters in determining the outcome of a drive release. We found that drive efficiency, fitness cost, and resistance rates were the most important factors in determining the outcome of a drive release, though several of the demographic parameters could also tip the scales between success and failure.

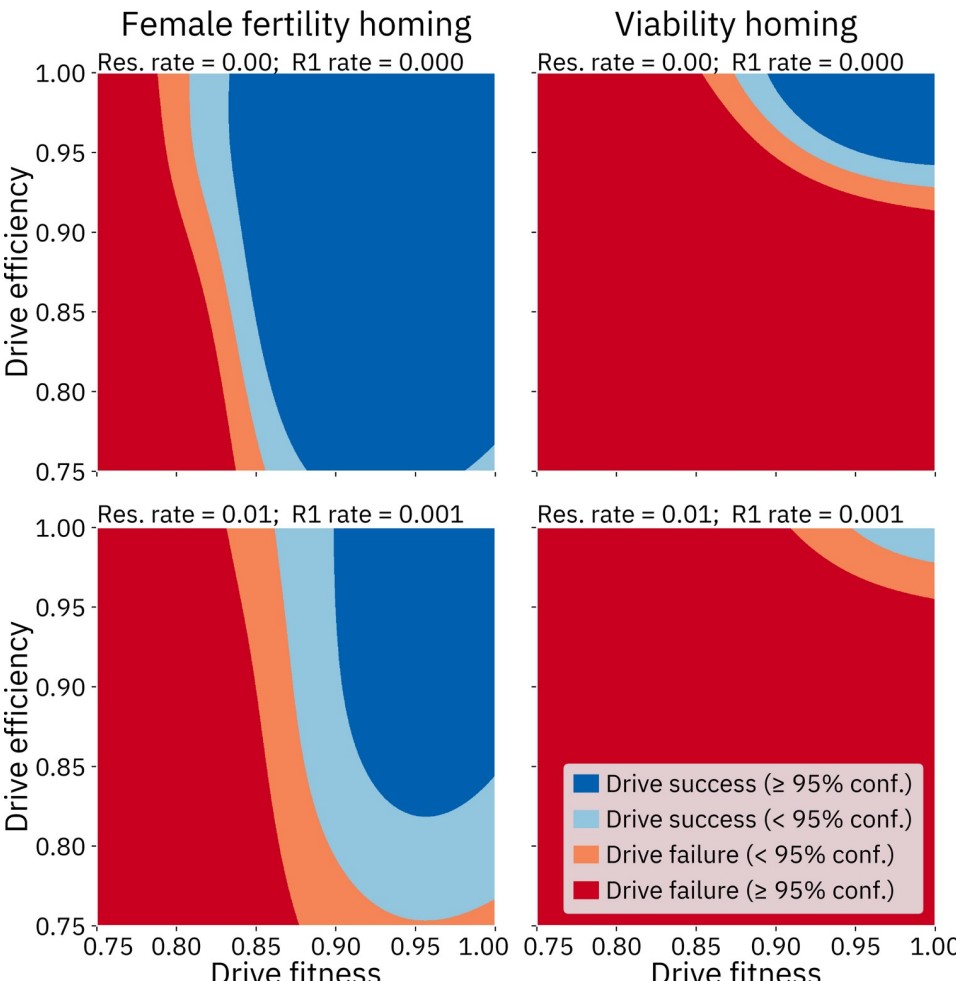

**Fig 14. Comparison of the two homing drives with varying drive fitness, efficiency, and resistance parameters.** Survival rate was set to 0.8, and other parameters were fixed at default values.

Our findings can help us develop strategies for improving the likelihood of successful population elimination. For example, our analysis revealed that any parameter choices that increase the mortality rate within the population also make it more likely that the drive will succeed. This suggests the possibility that the success of a drive could benefit greatly from the simultaneous application of traditional control measures, even if those measures are applied indiscriminately and have the possibility of eliminating drive carriers in addition to wild-type individuals [91,92]. To perform a cursory but explicit investigation of this possibility, we implemented a basic "traditional control rate" add-on parameter to our default population model. Each time step, a percentage of the population is eliminated according to this rate. Without a drive, upwards of 50% of the population must be removed in this way each time step in order to fully eliminate the population. Using a homing drive with a female fertility target with no fitness cost and a drive efficiency of 80%, population eradication can be achieved with an accompanying traditional control rate of 5%; the same drive with 70% efficiency can eliminate the population with a traditional control rate of 15% (without traditional control, such a drive required an efficiency of about 86% to reliably suppress the population). This

combined control strategy may prove to be an enabling factor in situations where neither strategy alone is sufficient.

Our model makes several abstractions that may be worth further consideration. For example, migration is modeled rather simplistically: individuals have a pre-defined chance to move in a random direction through a homogenous space. The odds of an individual migrating are not related to any prevailing conditions, and individuals do not necessarily end up in more favorable conditions than those they left. A great deal of complexity could be added to this system. Individuals might be given a higher chance to migrate when experiencing overcrowding or when they are unable to find mates. It would also seem more likely that individuals do not simply move to arbitrary destinations, but rather, tend to settle in areas where there are sufficient resources and potential mates available. The implementation of such resource-aware migratory behavior could change the dynamics and outcome of a simulated gene drive release.

Reproductive behavior could also be simulated in greater detail. In our model, each female reproduces during every time step. Yet, rats in the wild reproduce at a variable rate depending on the time of year and the amount of resources available. There was also no limit on the number of females that each male can reproduce with. In a healthy population at equilibrium, this may be a reasonable abstraction, but it might be inaccurate in our simulations of the Y-shredder gene drive. With an efficiency of 96% and otherwise default parameters, the Y-shredder does not reduce the size of the population at all, despite eventually reaching fixation. This means that, although the sex ratio of the population averages about one male to 50 females, the population size remains stable. As much as they may be a strongly *r*-selected species, it is unclear if male rats in wild populations would be capable of such feats of fertility.

Another abstraction in our model was a simplified version of gene drive dynamics. Future studies could implement a more advanced framework, including more detailed treatment of Cas9 cleavage and resistance sources, and multiplexed gRNA targets [45]. Additionally, while the three drives we considered in this study can rapidly spread in a population, they are all self-sustaining drives and could thus easily spread into non-target populations even under low rates of migration. Threshold-dependent drives such as CRISPR-based toxin-antidote systems [38,93] or tethered drives [94,95] have been proposed as a potential solution to this problem, and require much higher rates of migration in order to invade and spread into a non-target population. It may be interesting for future studies to extend our model to such "confineable" gene drive strategies.

## Gaussian process meta-model

The use of a GP model as an efficient and accurate surrogate model allowed us to conduct a comprehensive exploration of the parameter space of our underlying simulation model, which would have been prohibitively computationally expensive to perform on the simulation model directly. This means that future studies using this technique can use increasingly detailed simulation models without fear of long runtimes.

On the other hand, a GP model is only an approximator of the underlying model, and there may be situations when this is undesirable. For example, if an actual release candidate for a suppression gene drive were engineered, it may be preferable to create an extremely detailed simulation model where drive and species demographic characteristics are bounded to narrow intervals based on a careful analysis of the candidate drive and the targeted population. Such a model might then be most suitably analyzed without any approximation layer as it would require only a limited number of (albeit lengthy) evaluations to assess.

In the current study, the GP models we developed provide insights into the requirements for a gene drive to successfully eliminate an invasive rat population in a wide variety of

demographic contexts. This type of analysis could guide the engineering of gene drive systems by informing scientists of the minimum thresholds that a successful drive candidate must meet. However, this represents only the most basic application of a meta-model. A GP model can be trained on any output of the underlying model (or indeed, all of them), rather than just being used to predict whether or not suppression will occur. Utilizing additional outputs from the underlying model could enable a meta-model to answer many more questions about the system. For example, spatial distribution statistics could be gathered from each simulation [37] so that a meta-model is able to detect the conditions under which chasing occurs within the system. These statistics could also be combined with a suite of other model outputs in order to allow a complete mapping of all possible modes of drive failure and success.

Numerous changes could be made to the methods used to create our GP models that may result in even greater accuracy. To begin with, the meta-model framework was not altered as additional adaptive datapoints were added to the training set. For example, we elected to use the Matérn kernel after experimentation found it to give the best results with the initial training sets which consisted of only 1000 points. However, our final training set is much less sparse, and the models might now perform better with different kernels. We found that the Matérn kernel still outperforms the frequently used radial basis function kernel, but our testing of alternative kernels for the final models was not exhaustive.

Next, our training method was model agnostic. Our initial training set consisted of points from a Latin hypercube distribution, which is not necessarily the best choice for providing the model with the clearest understanding of which input factors contribute to variance in the output. Adaptively sampled points did a good job of improving the model, but improvements to the method of selecting new points could result in model accuracy improving more rapidly and to a greater degree. We anticipate that refinements to training techniques could provide substantial increases in accuracy in future studies.

Finally, the training process for each of our GP models was entirely separate. Although gene drive systems can behave very differently from one another, it might be reasonable to start with an assumption that parameters behave similarly across drives, and to proceed to learn the differences between systems from that starting point. A highly accurate GP model (such as our model for the homing drive with a female fertility target) could be used as a prior for models of other drives. Doing so would likely boost the accuracy of models created for drives with very small areas of success, which tend to have trouble identifying these areas of success to begin with, and thus tend to suffer from a higher incidence of false negatives. The strategy of using other models as a prior distribution may be even more appropriate in the context of modeling large families of similar drives, such as toxin-antidote systems [38,93,96].

## Sensitivity analysis

The rapid evaluation speed of the GP meta-models unlocks the ability to conduct variance-based sensitivity analyses of the system. Performing this type of analysis on our population model directly would require either vast computing resources or an impractical amount of time.

Sensitivity analysis can provide a more complete understanding of parameter importance and parameter interactions than traditional analyses. In an analysis wherein only two or three parameters are varied at a time, it can be difficult to decide which parameters to select. Even when a parameter is included in such an analysis, the same parameter might behave completely differently in other parts of the parameter space. For example, in the analysis in Fig 11, litter size does not seem to have much impact on the outcome of the homing drive targeting a female fertility gene. However, as indicated by the sensitivity analysis for that drive, litter

size is in fact important in other parts of the parameter space, as confirmed by a separate analysis (S11 Fig). We may have missed the importance of this parameter had we just performed the analysis in Fig 11.

Nevertheless, the sensitivity analyses we present here have a few limitations. First, since the sensitivity analyses were performed on the meta-models, they should only be relied on to the extent that those models are accurate representations of the underlying population model in the first place. The small confidence intervals from the plots should therefore be taken only as a starting point, with the actual confidence of the entire analysis being marginally worse for the homing drives without resistance, and moderately worse for the other models.

Next, while a sensitivity analysis can be performed on a black-box model, the results of the analysis are best understood in a light shed by an understanding of the implementation of the model. For instance, both homing drives, when modeled with resistance, displayed fairly little sensitivity to the r1 resistance rate. However, this does not mean that r1 resistance is less disruptive to drive performance than r2 resistance. The r1 resistance rate is implemented as a *relative* rate: the formation of r1 alleles depends on both that rate as well as on the overall resistance rate. This implementation causes the sensitivity index of the r1 parameter to be bounded by the sensitivity index of the overall resistance parameter. Further, the conversion rate of the drive is also partially dependent on the overall resistance rate parameter, since drive conversion cannot occur if a resistance allele forms first. This interaction increases the sensitivity of the model to changes in the overall resistance parameter. Awareness of these implementation choices suggests that r1 resistance is a more important consideration for these drives than the sensitivity indices at first indicate.

Finally, it should be noted that the sensitivity indices show the contributions of the parameters to variance in the output, which is not quite the same as the degree to which the parameters affect the drive. For example, the sensitivity analyses indicate that resistance is an important parameter for the homing drive with a fertility target, but it is not as important for the drive with a viability target. This seems to suggest that the homing drive with a viability target is somehow a more resistance-tolerant system. Yet, the analysis in Fig 14 indicates that this is not the case. The reason for this seeming contradiction is that, while resistance has a great deal of effect on the drive, the drive has a relatively smaller area of success in the first place, meaning resistance has a much smaller absolute effect on the model. In the panel of Fig 14 showing the female fertility targeting homing drive without resistance, success is predicted in 70% of the depicted area; for the viability targeting drive, success is predicted in only about 10% of the depicted area (these percentages closely match those found in the Latin hypercube sampling of the entire *n*-dimensional volume; see S1 Table). In the lower panels with resistance, these areas shrink to 45% and 1%, respectively. In this case, resistance has a larger relative impact on the viability targeting drive, but because of the relatively smaller area of success for that drive, it has a smaller impact on the model in terms of the absolute number of predictions that change from success to failure because of resistance.

The sensitivity analysis that includes resistance indicates that drive efficiency is far and away the largest contributor to variance in the model, not because the drive is especially tolerant of resistance, but because even a small amount of resistance can only be overcome by a drive with a very high efficiency. Thus, while the sensitivity indices of the resistance parameters are not large, the mere inclusion of those parameters in the model represents a large change in the dynamics of the system. The same is likely true of the spatial parameters: while many of these parameters have relatively low sensitivity indices, it is likely that the mere inclusion of spatial factors causes a paradigmatic shift in the outcome space of the model as compared to what might be expected of a panmictic model.

## Conclusion

Our model shows that a gene drive with a sufficiently high efficiency and a low fitness cost should be capable of eliminating island populations of invasive rodents under a variety of demographic assumptions, so long as resistance can be kept to minimal levels. As gene drive technology continues to develop, modeling approaches must be developed concurrently so that we can accurately predict the outcome of a drive release before any real-world application of the technology. The machine learning framework we present herein allows efficient and exhaustive analysis of population models, even models that are difficult to evaluate, which will in turn allow the development of models that make increasingly few abstractions. More generally, this technique offers great promise for modeling gene drive as well as other complex evolutionary systems.

## Supporting information

**S1 Fig. Gaussian Process model quality.** Precision and recall are shown for each of the Gaussian process models used in this study. Each model was evaluated against the Latin Hypercube test set prepared from the modeled drive.
(TIF)

**S2 Fig. Model comparison, female fertility homing drive with resistance.** Resistance was set to 0.01 and relative R1 resistance rate was set to 0.001. Other parameters are fixed at default values. Black, gray, or white square dots show the results from actual simulations, each denoting the result of twenty simulations.
(TIF)

**S3 Fig. Model comparison, viability homing drive without resistance.** Survival rate was set to 0.8, and other parameters were fixed at default values. Black, gray, or white square dots show the results from actual simulations, each denoting the result of twenty simulations.
(TIF)

**S4 Fig. Model comparison, viability homing drive with resistance.** Resistance was set to 0.01, relative R1 resistance rate was set to 0.001, and survival rate was set to 0.8. Other parameters were fixed at default values. Black, gray, or white square dots show the results from actual simulations, each denoting the result of twenty simulations.
(TIF)

**S5 Fig. Model comparison, Y-shredder.** Survival rate was set to 0.8, and other parameters were fixed at default values. Black, gray, or white square dots show the results from actual simulations, each denoting the result of twenty simulations.
(TIF)

**S6 Fig. Second order effects for the homing drive targeting a female fertility gene, modeled without resistance.** Second order effects describe the pairwise synergies of two parameters. Only the 12 largest effects are shown.
(TIF)

**S7 Fig. Second order effects for the homing drive targeting a female fertility gene, modeled with resistance.** Second order effects describe the pairwise synergies of two parameters. Only the 12 largest effects are shown.
(TIF)

**S8 Fig. Second order effects for the homing drive with a viability target, modeled without resistance.** Second order effects describe the pairwise synergies of two parameters. Only the 12

largest effects are shown.
(TIF)

**S9 Fig. Second order effects for the homing drive with a viability target, modeled with resistance.** Second order effects describe the pairwise synergies of two parameters. Only the 12 largest effects are shown.
(TIF)

**S10 Fig. Second order effects for the Y-shredder drive.** Second order effects describe the pairwise synergies of two parameters. Only the 12 largest effects are shown.
(TIF)

**S11 Fig. Female fertility homing drive with varying litter size and drive efficiency.** Other parameters are fixed at default values. An initial analysis (in Fig 11) suggested that litter size was not important for this drive; however, this analysis confirms that litter size can make the difference between success and failure of the drive, as indicated by sensitivity analyses (Figs 8 and 9).
(TIF)

**S12 Fig. Viability homing drive with varying fitness, efficiency, and survival rate.** Other parameters are fixed at default values.
(TIF)

**S1 Table. Additional Model Comparisons.** The GP model for the female fertility homing drive including resistance was also tested against the test set prepared for the model for the female fertility homing drive without resistance. A similar test was performed on the model for the viability homing drive including resistance. The range for the drive fitness and efficiency parameters is [0.75, 1] in the models with resistance and [0.5, 1] in the models without resistance. Thus, this test was performed using only the subset of the test set where fitness and efficiency were in the range [0.75, 1]. For comparison purposes, the models without resistance were tested against this subset of the testing sets as well, and the results are included above. It appears that in three out of four cases, the model with resistance performs very nearly as well, despite having trained on a parameter space that is two dimensions larger. In the case of the composite model for the viability homing drive with resistance, the model quality reduction is larger.
(XLSX)

## Author Contributions

**Conceptualization:** Samuel E. Champer, Nathan Oakes, Philipp W. Messer.

**Funding acquisition:** Pablo García-Díaz, Philipp W. Messer.

**Investigation:** Samuel E. Champer, Nathan Oakes, Ronin Sharma, Pablo García-Díaz, Jackson Champer, Philipp W. Messer.

**Methodology:** Samuel E. Champer, Nathan Oakes, Ronin Sharma, Pablo García-Díaz, Jackson Champer, Philipp W. Messer.

**Project administration:** Philipp W. Messer.

**Supervision:** Philipp W. Messer.

**Visualization:** Samuel E. Champer.

**Writing – original draft:** Samuel E. Champer, Nathan Oakes, Philipp W. Messer.

**Writing – review & editing:** Samuel E. Champer, Nathan Oakes, Ronin Sharma, Pablo García-Díaz, Jackson Champer, Philipp W. Messer.

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
