## [Decision Letter · Decision Letter 0]

15 Jun 2021

Dear Prof. Messer,

Thank you very much for submitting your manuscript "Modeling CRISPR gene drives for suppression of invasive rodents" for consideration at PLOS Computational Biology.

As with all papers reviewed by the journal, your manuscript was reviewed by members of the editorial board and by several independent reviewers. In light of the reviews (below this email), we would like to invite the resubmission of a significantly-revised version that takes into account the reviewers' comments.

We cannot make any decision about publication until we have seen the revised manuscript and your response to the reviewers' comments. Your revised manuscript is also likely to be sent to reviewers for further evaluation.

Sincerely,

Benjamin Muir Althouse

Associate Editor

PLOS Computational Biology

Natalia Komarova

Deputy Editor

PLOS Computational Biology

Reviewer's Responses to Questions

**Comments to the Authors:**

Reviewer #1: In this article, Champer et al. have used an existing population genetic framework to simulate the impact of suppression gene drive strategies in eliminating rat species in an island setting. Given the large parameter space associated with individual-based dynamical systems they use a Gaussian process model to learn important parameters and their associated value ranges that would lead to gene drive success. They also perform sensitivity analyses on the model albeit the GP one. This article contains a number of details but I think the authors could do a better job of highlighting the take home points and the major contribution of their work. I am unclear if this work aims to evaluate target profiles of rat gene drives, how to develop an emulator, model a specific physical context, none of these, or all of these? The parameter space is large for these models, however, a number of these parameters ought to be tied to real world data (such as migration rates as well as gene drive parameters), which should limit the parameter space greatly. In fact, this has been proven multiple times with gene drive models of mosquitoes and other insects. So as a major suggestion, I request the authors to tighten up the language to communicate their goals and what they did to achieve them more clearly. Second, I would like to see more grounding in data. Rat migration and reproduction are well studied in literature and making assumptions about these characteristics without grounding them in data does not engender confidence in the reader. More specific comments are as follows:

Impact of migration on fixation

Introduction:

1. In the section on spatially abstract models, the following two would be very relevant to include because they are models explicitly created to simulate genetics and gene drive models as opposed to spatial models of disease transmission that then included gene drive as an add on:

Sanchez et al. (2019): https://besjournals.onlinelibrary.wiley.com/doi/full/10.1111/2041-210X.13318

Selvaraj et al. (2020):

https://journals.plos.org/ploscompbiol/article?id=10.1371/journal.pcbi.1008121

Methods:

1. "Time steps in the simulation are equivalent to one breeding cycle, which in reality can range from approximately one to six months depending on the availability of resources, as well as the climate and time of year" Please mention how this compares to the average lifespan of the species being modeled.

2. "The simulation is then run for an additional 500 time steps or until the population is eliminated." Does the 500 time steps imply there are scenarios where elimination is not achieved?

3. Why was a Gaussian curve chosen for competition exerted? Is this fit to some data showing loss in attraction or purely an assumption? Please clarify.

4. Are there other fitness costs apart from fertility and viability?

Gaussian Process:

1. Are 20 replicates at each simulated point enough to estimate stochastic variation in the model?

2. "This value is composed of two terms: one denoting the speed at which suppression occurred (if it occurred) consisting of the last generation in which there were living individuals divided by the total number of generations simulated (500)" - Isn't this true for the denominator only if the simulations haven't eliminated before 500 steps?

3. If the suppression model can produce biased results because of sharp transitions, why use the suppression model at all? Isn't it better to just stick to the composite model?

Results:

Population Model:

1. What exactly is the discrepancy in the density calibration model? This feels a bit hand wavy and I'd appreciate more detail here because the point of an agent based model is to be able to track individuals explicitly and not having exact control over the population size does not make sense to me.

2."This correlates fairly well with the default maximum competition distance of 75 meters". Is this some standard distance for max competition?

3."The transition between invariable failure and invariable success occurs rather abruptly as efficiency is increased beyond a threshold". Is this a feature of model dynamics? Why does this happen? The authors addressed how they got around this while using the Gaussian Process model but I still don't understand why this is happening.

4. In figure 2, I see there are simulations that eliminated at 490 time steps. Are there simulations that would have eliminated at 510 time steps? Why is 500 chosen as the maximum simulation time? The definition for elimination/persistence has to be more rigorous than an arbitrarily chosen time step.

5. Where exactly are the genetically modified rats released? Figure 3 suggests a random release?

6. I'm still confused about the two GP models. In the methods, the authors state the composite model would offer a solution around abrupt transitions but it looks like the accuracy is a lot lower than the suppression only GP model. So what is the way to go then?

7. "it appears that the suppression rate model often has areas of more exaggerated curvature than the composite model". Is this with reference to figure 5? I'm not sure I see the exaggerated curvature. What do the authors mean here?

Reviewer #2: See attached review.

**Have the authors made all data and (if applicable) computational code underlying the findings in their manuscript fully available?**

Reviewer #1: Yes

Reviewer #2: Yes

PLOS authors have the option to publish the peer review history of their article (what does this mean?). If published, this will include your full peer review and any attached files.

Reviewer #1: No

Reviewer #2: No
---

## [Decision Letter · Decision Letter 1]

18 Nov 2021

Dear Prof. Messer,

We are pleased to inform you that your manuscript 'Modeling CRISPR gene drives for suppression of invasive rodents using a supervised machine learning framework' has been provisionally accepted for publication in PLOS Computational Biology.

Best regards,

Benjamin Althouse

Associate Editor

PLOS Computational Biology

Natalia Komarova

Deputy Editor

PLOS Computational Biology

Reviewer's Responses to Questions

**Comments to the Authors:**

Reviewer #1: The authors have sufficiently addressed my comments. I recommend this article proceed in the publication process.

Reviewer #2: The authors have addressed all concerns outlined in my initial review, so I advise to publish this manuscript as is.

**Have the authors made all data and (if applicable) computational code underlying the findings in their manuscript fully available?**

Reviewer #1: None

Reviewer #2: Yes

PLOS authors have the option to publish the peer review history of their article (what does this mean?). If published, this will include your full peer review and any attached files.

Reviewer #1: No

Reviewer #2: No

---

## [Editor Report · Acceptance letter]

9 Dec 2021

PCOMPBIOL-D-21-00773R1 

Modeling CRISPR gene drives for suppression of invasive rodents using a supervised machine learning framework

Dear Dr Messer,

I am pleased to inform you that your manuscript has been formally accepted for publication in PLOS Computational Biology. Your manuscript is now with our production department and you will be notified of the publication date in due course.

With kind regards,

Zsofia Freund
